# Scalable Hierarchical Self-Attention with Learnable Hierarchy for Long-Range Interactions

**Thuan Trang**[*]                                                              *thuantna@fsoft.com.vn*
*FPT Software AI Center*

**Nhat Khang Ngo**[*]                                                           *khangnn4@fpt.com*
*FPT Software AI Center*

**Hugo Sonnery**[*]                                                            *hugo.sonnery@mail.mcgill.ca*
*McGill University*
*Mila - Quebec AI Institute*

**Thieu N. Vo**                                                                *vongocthieu@tdtu.edu.vn*
*Ton Duc Thang University*
*FPT Software AI Center*

**Siamak Ravanbakhsh**                                                         *siamak@cs.mcgill.ca*
*McGill University*
*Mila - Quebec AI Institute*

**Truong Son Hy**[†]                                                           *TruongSon.Hy@indstate.edu*
*Indiana State University*
*FPT Software AI Center*

**Reviewed on OpenReview:** *https://openreview.net/forum?id=qH4YFMyhce*

## Abstract

Self-attention models have made great strides toward accurately modeling a wide array of data modalities, including, more recently, graph-structured data. This paper demonstrates that adaptive hierarchical attention can go a long way toward successfully applying transformers to graphs. Our proposed model Sequoia provides a powerful inductive bias towards long-range interaction modeling, leading to better generalization. We propose an end-to-end mechanism for a data-dependent construction of a hierarchy which in turn guides the self-attention mechanism. Using adaptive hierarchy provides a natural pathway toward sparse attention by constraining node-to-node interactions with the immediate family of each node in the hierarchy (e.g., parent, children, and siblings). This in turn dramatically reduces the computational complexity of a self-attention layer from quadratic to log-linear in terms of the input size while maintaining or sometimes even surpassing the standard transformer's ability to model long-range dependencies across the entire input. Experimentally, we report state-of-the-art performance on long-range graph benchmarks while remaining computationally efficient. Moving beyond graphs, we also display competitive performance on long-range sequence modeling, point-clouds classification, and segmentation when using a fixed hierarchy. Our source code is publicly available at `https://github.com/HySonLab/HierAttention`.

---

[*]These authors contributed equally to this work
[†]Correspondence to TruongSon.Hy@indstate.edu

## 1 Introduction

Transformers (Vaswani et al., 2017) are powerful models that demonstrated superior performance in various input modalities, ranging from natural language processing to video understanding. Self-attention has also proved to be an effective tool for handling long-range dependencies, although its efficiency suffers due it its **quadratic complexity** in the input size. Therefore, a considerable body of literature on efficient transformers is dedicated to scaling the vanilla transformer with multiple approaches to resolving their scaling/performance trade-off. Strategies to make them more compute-efficient include approximating the attention matrix with sparsity patterns (Ho et al., 2020; Beltagy et al., 2020a; Zaheer et al., 2020a), clustering before computing attention (Roy et al., 2021; Kitaev et al., 2020b), low-rank estimation (Wang et al., 2020; Xiong et al., 2021) and better memory I/O (Dao et al., 2022).

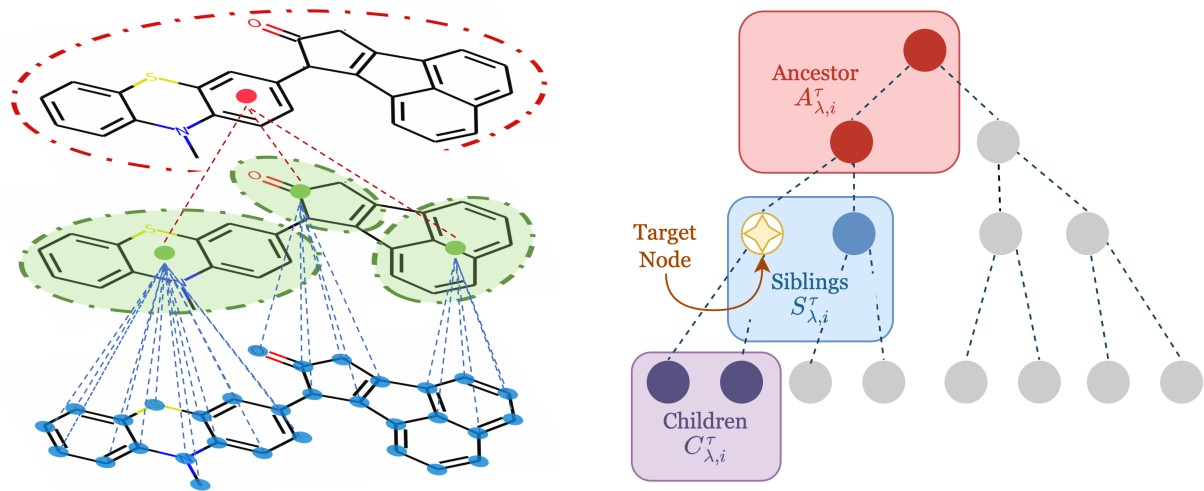

Figure 1: (**Left**) We adaptively partition the graph to form a hierarchy. This incremental partitioning at each level is produced by message passing on the graph at a lower level and the use of Gumbel-max for hard clustering. (**Right**) This adaptive tree forms the basis for sparse self-attention in a single self-attention layer.

In this work, we focus on devising a self-attention mechanism that could enable **long-range interaction modeling on graph-structured data**. Most current efficient transformers either sparsify the attention matrix using random patterns or a distance metric derived from the tokens' relative positions in a sequence. Our initial motivation coincides with a similar idea of skipping superfluous interactions to capture long-range interactions under a fixed computational budget (Choromanski et al., 2021; Katharopoulos et al., 2020). More precisely, our model Sequoia decreases the computational cost by **exploiting an adaptive task-informed hierarchical clustering of the graph rather than using a prespecified interaction pattern**. As a result, our method not only requires lower resources but also surpasses the performance of standard attention due to additional information about the data structure provided by the **learnable hierarchy**. Furthermore, by propagating the information bottom-up and top-down through the tree built atop the input graph, our method guarantees local and global information sharing. The resulting adaptive hierarchical self-attention layer can be used as a **building block for geometric deep learning using transformers**.

Our model is comprised of two parts : (i) **computational tree learning** that forms the basis for (ii) **hierarchical attention** mechanism. Our work attempts to cover long-range datasets with various structures, including what we categorize as ordered, semi-ordered, and unordered data. This distinction is related to the way the hierarchy is constructed in each case: for ordered data structures, such as images and long sequences, a hierarchy may be naturally constructed from their regular grid. More concretely, a sequence or image is divided into nested sub-sequence or image patches, with decreasing resolution and increasing receptive field at each level of the hierarchy. For semi-ordered structures such as point clouds, such a grid

structure can be "imposed", for example, using nested voxelization (Liu et al., 2019). However, graphs, as an example of an unordered structure, arguably lack a natural hierarchy, and we learn the hierarchy for each data instance. Due to the effectiveness of adaptive hierarchy for graphs, we focus our discussions on graphs; discussion and further results on long-range sequences and point clouds are moved to the appendix.

In short, the main contributions of this paper are:

- An **efficient hierarchical self-attention scheme** with $O(n \log(n))$ space and time complexity in terms of the sequence length / the number of points / the graph size.
- A **data-driven approach to constructing a tree** as a guideline for the attention mechanism for long-range graphs.
- Extensive experiments on **long-range graphs understanding** as well as long-range sequence modelling and point clouds with state-of-the-art or competitive results along with ablation studies to demonstrate the effectiveness and efficacy of our method.

## 2 Related work

Our contribution is related to a wide range of literature on deep learning on graphs, and in particular application of transformer architectures to graphs, as well as the line of research of efficient transformers. Below, we present a concise overview and further motivate our approach in the light of related literature:

**Learning on graphs**  Graph neural networks based on **message-passing** scheme (Scarselli et al., 2009; Gilmer et al., 2017; Kipf & Welling, 2016; Xu et al., 2019; Corso et al., 2020; Hamilton et al., 2017; Veličković, 2022) are currently the most widely adopted architectures for learning on graphs. By imposing a sparsity pattern derived from the input graph on the computation graph, this type of neural architecture benefits from **linear complexity in the input size**. While efficient and often competitive, the message-passing scheme based on propagating and aggregating information among local neighbourhoods possesses well-known limitations such as too much **focus on locality** (Morris et al., 2019; Xu et al., 2019), **over-squashing** (Alon & Yahav, 2021; Topping et al., 2022) and **over-smoothing** (Chen et al., 2020a; Oono & Suzuki, 2020), preventing them from dealing with long-range graphs (Dwivedi et al., 2022b) or capturing multiscale structures (Hy & Kondor, 2023; Ying et al., 2018). In particular, several previous works on graph learning adaptively construct a hierarchy using differentiable pooling (Gao & Ji, 2019; Knyazev et al., 2019; Khasahmadi et al., 2020; Lee et al., 2019b; Ranjan et al., 2019). Their main distraction with our work is that they construct a soft cluster assignment, which does not help with computational complexity when working with transformers. Moreover, pooling operations only enable information sharing in a bottom-up direction, while our hierarchical self-attention, enables information sharing between parents, children and siblings.

Recently, transformers built upon the self-attention mechanism, which takes into account all pairwise interactions between nodes, has been employed to alleviate these issues (Müller et al., 2023).

**Transformers for graphs**  Previous studies have adapted conventional transformers to work on graph domains by introducing novelties, such as graph feature encoding schemes (Dwivedi & Bresson, 2020a; Rampasek et al., 2022; Ying et al., 2021), spectral attention (Kreuzer et al., 2021), graph tokenization (Kim et al., 2022a), structural information extraction (Chen et al., 2022; Rong et al., 2020), relative positional encoding (Mialon et al., 2021), and attention on meta-paths (Yun et al., 2019). These techniques enable transformers to display good performance on a wide range of small graph benchmarks (Hu et al., 2020; Sterling & Irwin, 2015). However, extensions of vanilla transformers, which retain quadratic complexity, encounter challenges in terms of memory and execution time when the graphs become larger (Dwivedi et al., 2022b). **Our work aims at bridging the gap between graph transformers and message-passing networks**. Building graph hierarchies enables us to operate local attention at each level and achieve global attention by aggregating and updating information between the levels. This, as a result, helps with the communication between distant nodes, while benefiting from a sub-quadratic complexity.

**Efficient transformers for modeling long-range dependencies**  To ease the quadratic complexity of vanilla transformer in modeling long sequences, several mechanisms were proposed to simplify the computa-

tion of the attention matrix as summarized in survey (Lin et al., 2022). Reformer (Kitaev et al., 2020a) is a content-based sparse attention transformer that matches tokens together into buckets using locality-sensitive hashing before applying local attention. Others employ learnable sparsity patterns, like Sinkhorn transformers (Tay et al., 2020a) which induce sparsity by sorting the keys and values matrices such that local heuristics can be applied in the computation of the scaled-dot-product. Most efficient transformers (Guo et al., 2019), (Beltagy et al., 2020b; Ravula et al., 2020; Zaheer et al., 2020b) resort to similar sparsity heuristics to get rid of the bulk of the computations in the dot-product attention in a structured way, allowing for efficient implementation without custom CUDA kernels. They often rely on local information processing, only selecting a few tokens as "global tokens" which will attend to every token and can be attended to by all other tokens. Further away from sparsity-based approaches, Transformer-XL (Dai et al., 2019) is a memory-based transformer that adds recurrent connections across segments by reusing the hidden states of the model while processing tokens of the previous segment. The work closest to ours is BP Transformer (Ye et al., 2019), which uses binary partitioning of a sequence in order to integrate information from long-term context in a fine-to-coarse fashion. Our model relies on sequential local message-passing updates in a tree, as opposed to BP Transformer, which achieves global information sharing by computing intermediate nodes' embeddings in parallel directly from their leaf nodes instead of their direct children. Also, it relies on the sequential ordering of the tokens to construct the dynamic hierarchy, which is impossible to do on graph-structured data for which a natural monotonous ordering does not always exist.

We argue that models from the efficient transformers literature outlined above are not suitable for graph-structured data for at least three reasons:

1. Their approach to sparsification may rely on 1D distance between tokens (e.g. convolutional-sense locality) and thus **cannot be applied to other input modalities** than sequences.

2. Their efficient implementation on GPUs often **requires custom CUDA kernels**, which limits the widespread availability of such models.

3. Most importantly, they often accomplish sparsification via mathematical tricks which may or may not **constitute a natural or motivated inductive bias** towards better generalization.

As a consequence, **vanilla attention remains a strong state-of-the-art** general-purpose model in terms of task performance for a wide array of input modalities. We believe that the main limitation towards better task performance of a general-purpose model is the ability to retrieve specific information from long-range interactions, which is lost in the "noise" of the large sum over input tokens performed by attention pooling over large input data. In this paper, we aim not only to perform on par with vanilla attention on designated benchmarks but also to **outperform classical attention in terms of task performance** thanks to the powerful inductive bias provided by adaptive hierarchical attention.

## 3 Method

Overall, our proposed methods comprise two interacting components: (i) **learning a hierarchical latent structure over the input tokens** and (ii) applying a **sparse attention scheme over the recursive $k$-ary tree** computed in (i) in order to update tokens' embeddings.

The $k$-ary tree is built over the input tokens, which correspond to leaf nodes. The other intermediate levels of the tree up until the root node will henceforth be denoted *virtual nodes*. We use $X \in \mathbb{R}^{n \times d}$ to denote the embedding of input tokens or node features, and $\mathcal{E} \in \mathbb{R}^{n \times n}$ for the weighted adjacency matrix – that is $\mathcal{E}_{ij} > 0$ implies an edge between node $v_i$ and $v_j$ with weight $\mathcal{E}_{ij}$. We use $\tau = \tau(X, \mathcal{E})$ for a $k$-ary tree, where $n = \sum_{\lambda=0}^{\Lambda-1} n_\lambda$ is the total number of nodes in the tree, including both leaf nodes and internal or virtual nodes. Here, $\lambda \in \{0, \ldots, \Lambda - 1\}$ indexes the tree level, where $\lambda = 0$, $\lambda = \Lambda - 1$ indicates leaf level and root level, respectively. Using this indexing, $X_\lambda \in \mathbb{R}^{n_\lambda \times d}$ is the embeddings matrix of the nodes at level $\lambda$. Similarly, $\mathcal{E}_\lambda \in \mathbb{R}^{n_\lambda \times n_\lambda}$ corresponds to the adjacency matrix at that level. Finally, we use $I_\lambda \subset \{0, \ldots, n\}$ for node indices in a given layer.

Next, we elaborate on two components of our model, adaptive construction of hierarchy and sparse self-attention guided by the resulting tree.

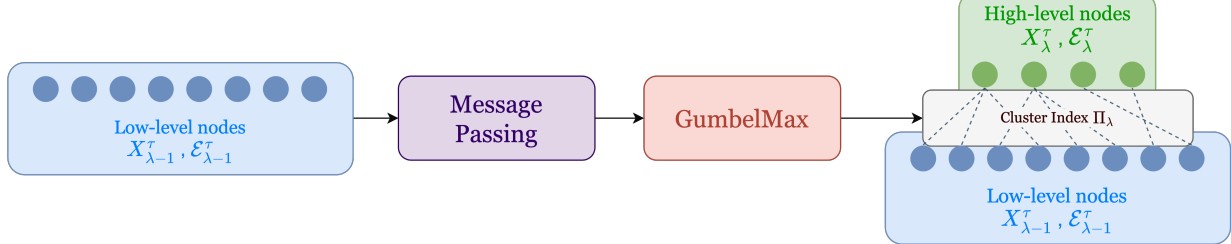

Figure 2: Our proposed Cluster Learning Block. In this block, the features of low-level nodes are used to learn the cluster presented as virtual high-level nodes.

### 3.1 Adaptive Hierarchy

Although Sequoia's $k$-ary tree can be pre-constructed using a partitioning algorithm designed especially for a given input domain, we provide an approach to **learn the clustering automatically**. In contrast to previous works (Gao & Ji, 2019; Knyazev et al., 2019; Khasahmadi et al., 2020; Lee et al., 2019b; Ranjan et al., 2019) that learn a soft clustering assignment for nodes, ours learns to partition the inputs into mutually exclusive clusters and coarsen into the upper-level resolution in the hierarchy, which enables inter-level and intra-level attention at different resolutions.

A hard clustering of $n$ nodes into $k$ clusters is a mapping $\pi : \{1, .., n\} \to \{1, ..., k\}$ where $\pi(i) = j$ if node $i$ is assigned to cluster $j$. The clustering is represented by an assignment matrix $\Pi \in \{0, 1\}^{n \times k}$, where $\Pi_{i, \pi(j)} = 1$. Since we require this hard assignment to be the output of a neural network, we relax $\Pi$ to $P \in (0, 1)^{n \times k}$, a row-stochastic matrix representing a distribution over clusters. We then sample from this distribution to produce a cluster assignment. To make this process differentiable, we employ the Gumbel-max trick (Gumbel, 1954; Maddison et al., 2014; Jang et al., 2016) that provides a simple implementation and efficient way to define a categorical distribution $\pi$ as:

$$\Pi_i = \text{one-hot}\big(\text{argmax}\big[g_{i,k} + \log P_{i,k}\big]\big),$$

where $P_i = [P_{i,1}, ..., P_{i,k}]$, is a vector of class probabilities computed by Softmax, and $g_{i,j} \sim \text{Gumbel}(0, 1)$ are i.i.d samples from Gumbel distribution.

To construct the hierarchy in a bottom-up fashion, starting from leaf nodes we use a Message-Passing Neural Network(MPNN) on the corresponding graph $(X_\lambda, \mathcal{E}_\lambda)$ to compute the probability matrix $P$ and apply the GumbelMax function to derive $\Pi$, and therefore the layer above in the hierarchy:

$$P_\lambda = \text{MPNN}(X_{\lambda-1}, \mathcal{E}_{\lambda-1}),$$
$$\Pi_\lambda = \text{GumbelMax}(P_\lambda).$$

As $\Pi_\lambda$ is differentiable, the parameters of MPNN can be updated via backpropagation, resulting in an effective end-to-end method for learning to cluster. For each level $\lambda$, corresponding node indices $I_\lambda$ can be easily retrieved from cluster memberships $\Pi_\lambda$ by converting from one-hot vector to index.

After identifying the cluster memberships in the layer above in the hierarchy, we use the membership to aggregate the features of lower nodes and edges to construct higher-level node and edge features:

$$X_\lambda = \Pi^\top X_{\lambda-1}, \quad \mathcal{E}_\lambda = \Pi^\top \mathcal{E}_{\lambda-1} \Pi. \tag{1}$$

The interaction in the opposite direction is also straight-forward:

$$X_{\lambda-1} = \Pi X_\lambda. \tag{2}$$

We term these two operations as **Pooling**(Eq. 1) and **Expanding**(Eq. 2) layers.

In short, when forming a tree, the nodes and edges at a higher level are constructed based on the lower level. Specifically, after the input is fed into Adaptive Hierarchy, the block yields assignment matrix $\Pi$. Based on this matrix $\Pi$ and low-level nodes $X$ and edges $\mathcal{E}$, we can substitute them into Eq. 1 to obtain the higher-level nodes and edges, resulting in weighted graphs. Optionally, the equation can also applicable for edge attribute. However, in our implementation, for the simplicity, we keep the graph unweighted, which means we only consider whether two nodes are connected without considering the weight of the connection. The pseudocode for this process is presented in Alg. 1.

### 3.2 Hierarchical Self-Attention Block

Sequoia's hierarchical attention scheme attends to three different neighbourhoods based on the tree structure:

- Children $C_{\lambda,i}$ (all nodes $j \in I_{\lambda-1} \mid \text{Parent}[j] = i$).
- Siblings $S_{\lambda,i}$ (all nodes $j \in I_\lambda \mid \text{Parent}[j] = \text{Parent}[i]$).
- Ancestors $A_{\lambda,i}$ : (all nodes $j \in I_\lambda \cup \cdots \cup \in I_{\Lambda-1} \mid \underbrace{\text{Parent} \circ \cdots \circ \text{Parent}}_{\Lambda-\lambda \text{ times}} [i] = j$).

Our proposed attention scheme consists in applying:

$$\text{ATTN}(\cdot) = \mathcal{A}^{\text{ent}}(\cdot) = \phi\left(\mathcal{A}^{\text{c}}(\cdot), \mathcal{A}^{\text{s}}(\cdot), \mathcal{A}^{\text{a}}(\cdot)\right),$$

where $\phi$ an appropriate fusion layer mixing the updated embeddings of each node according to the three different attention types ( c - *children*, s - *siblings* and a - *ancestors*) on the node axis. Overall, for each node $i \in I_\lambda$, we first gathers three types of neighbors $j$ of target nodes $i$, including children $C_{\lambda,i}$, siblings $S_{\lambda,i}$, and ancestors $A_{\lambda,i}$ using the tree structure - illustrated in Figure 1. Then, we compute three different types of embeddings $\mathcal{A}^{\text{c}}$, $\mathcal{A}^{\text{s}}$, and $\mathcal{A}^{\text{a}}$ for the target node based on its attention scores to each type of neighbor. These resulting embeddings are aggregated using pooling to produce a unifying embedding $\mathcal{A}^{\text{ent}}$ for the target node.

More formally, let $Q_i$ be the query embedding associated with node $i$, and $(K_j, V_j)_{j \in \mathcal{V}}$ be the key and value embeddings associated with nodes $j$ in the neighbourhood, then the attention for children $\mathcal{A}^{\text{c}}_{\tau,\lambda,i}$, siblings $\mathcal{A}^{\text{s}}_{\tau,\lambda,i}$, ancestors $\mathcal{A}^{\text{a}}_{\tau,\lambda,i}$ of node $i$ at level $\lambda$ in $k$-ary tree $\tau$ is defined as:

$$\mathcal{A}^{\text{c}}_{\tau,\lambda,i}(X^\tau_\lambda, i) = \text{Attention}_{W^c_Q, W^c_K, W^c_V}\left[Q_i, (K_j, V_j)_{j \in C_{\lambda,i}}\right],$$

$$\mathcal{A}^{\text{s}}_{\tau,\lambda,i}(X^\tau_\lambda, i) = \text{Attention}_{W^s_Q, W^s_K, W^s_V}\left[Q_i, (K_j, V_j)_{j \in S_{\lambda,i}}\right],$$

$$\mathcal{A}^{\text{a}}_{\tau,\lambda,i}(X^\tau_\lambda, i) = \text{Attention}_{W^a_Q, W^a_K, W^a_V}\left[Q_i, (K_j, V_j)_{j \in A_{\lambda,i}}\right].$$

We finally fuse the updated embeddings proposed for node $i$ by each of the three attention types using simple average pooling, which can also be sum or max pooling. Here, we do not combine the embeddings of children, siblings and ancestors with the embedding of each node itself because they will be implicitly combined using residual connections mentioned in the next section.

$$\mathcal{A}^{\text{ent}}_{\tau,\lambda,i} = \phi_{\text{average}}[A^{\text{c}}_{\tau,\lambda,i}, A^{\text{s}}_{\tau,\lambda,i}, A^{\text{a}}_{\tau,\lambda,i}] = \frac{A^{\text{c}}_{\tau,\lambda,i} + A^{\text{s}}_{\tau\lambda,i} + A^{\text{a}}_{\tau,\lambda,i}}{3}.$$

### 3.3 Putting it all together

Initially, the cluster learning block is applied repeatedly $\Lambda - 2$ times to learn an $\Lambda$-layers tree. Here, the root node representation is given by pooling the $X_{\Lambda-2}$, without any "learning". The term "without any learning" means that the root node's embedding is not initialized as a learnable parameter that is updated via back-propagation, but is the aggregation of node embeddings of the low-level nodes. After constructing the tree structures level by level, a combination of message passing, hierarchical attention, and residual connections are used to refine node features, before pooling (Eq. 1) them to produce the features for the higher level; we call these steps *bottom-up blocks* (Alg. 2). By employing the bottom-up iteratively from the leaves up to

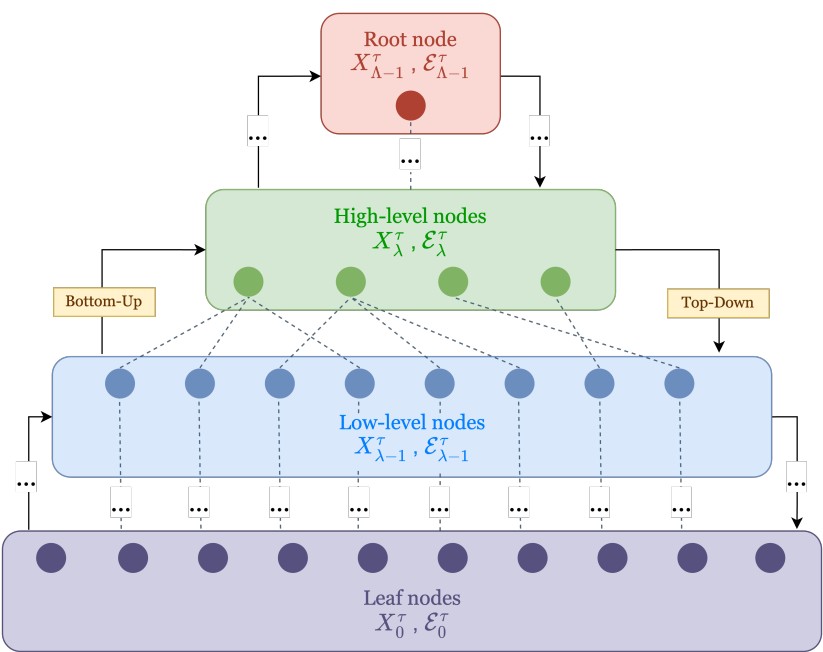

Figure 3: Our proposed Sequoia which mainly uses Bottom-Up and Top-Down block to aggregate local features from leaves to root and propagate global features in the opposite direction.

the root, the model can extract global information from local features. To propagate back the information to local features, a *top-down block* (Alg. 3) is used. In this block the feature from the higher level needs to be expanded to have the same shape as the lower one. Subsequently, we apply MLP on refined lower-level and then add both expanded (Eq. 2) and refined features together. The details of the combination of all mentioned blocks are presented in Alg. 4. This block is also applied several times until the information reaches the leaf level. We refer the readers to the Appendix for a time-complexity analysis.

---

**Algorithm 1:** Building Hierarchical Tree

---

**Input:** $X_{\lambda-1}, \mathcal{E}_{\lambda-1}$ ;       ▷ Inputs are nodes and edges of low-level in the hierarchy
**Output:** $X_\lambda, \mathcal{E}_\lambda$ ;       ▷ Outputs are nodes and edges of high-level in the hierarchy
$P_\lambda = \mathrm{MPNN}(X_{\lambda-1}, \mathcal{E}_{\lambda-1})$ ;       ▷ Apply the MPNN to compute the probability matrix $P$
$\Pi_\lambda = \mathrm{GumbelMax}(P_\lambda)$ ;    ▷ Apply the GumbelMax function to derive the assignment matrix $\Pi$
$X_\lambda = \Pi^\top X_{\lambda-1}$ ;       ▷ Compute the high-level nodes $X_\lambda$
$\mathcal{E}_\lambda = \Pi^\top \mathcal{E}_{\lambda-1} \Pi$ ;       ▷ Compute high-level edges $\mathcal{E}_\lambda$

---

**Algorithm 2:** Bottom-up Block

---

**Input:** $X_{\lambda-1}, \mathcal{E}_{\lambda-1}$ ;       ▷ Input is nodes and edges of low-level in the hierarchy
**Output:** $X_\lambda$ ;       ▷ Output is nodes of high-level in the hierarchy
$X_{local} = X_{\lambda-1} + \mathrm{MPNN}(X_{\lambda-1}, \mathcal{E}_{\lambda-1})$ ;       ▷ Apply the MPNN to compute the local features
$X_{global} = X_{\lambda-1} + \mathrm{ATTN}(X_{\lambda-1}))$ ;       ▷ Apply the ATTN to compute the global features
$X'_{\lambda-1} = X_{local} + X_{global}$ ;       ▷ Combine local and global features
$X_\lambda = \mathrm{POOLING}(X'_{\lambda-1})$ ;       ▷ Compute high-level nodes $X_\lambda$

---

---

**Algorithm 3:** Top-down Block

---

**Input:** $X_{\lambda-1}, X_\lambda, \mathcal{E}_\lambda$ ;  ▷ Input is nodes and edges of high-level in the hierarchy
**Output:** $X''_{\lambda-1}$ ;  ▷ Output is nodes of low-level in the hierarchy
$X_{local} = X_\lambda + \text{MPNN}(X_\lambda)$ ;  ▷ Apply the MPNN to compute the local features
$X_{global} = X_\lambda + \text{ATTN}(X_\lambda)$ ;  ▷ Apply the ATTN to compute the global features
$X'_\lambda = X_{\text{local}} + X_{\text{global}}$ ;  ▷ Combine local and global features
$X'_{\lambda-1} = \text{UNPOOLING}(X'_\lambda)$ ;  ▷ Compute new low-level nodes
$X''_{\lambda-1} = X_{\lambda-1} + X'_{\lambda-1}$ ;  ▷ Combine old and new low-level nodes to generate $X''_{\lambda-1}$

---

---

**Algorithm 4:** Entire Network

---

**Input:** $X_0, \mathcal{E}_0$ ;  ▷ Input is leaf nodes of the hierarchy
**Output:** $X'$ ;  ▷ Output is nodes of all levels in the hierarchy
**for** *(i = 0, i < λ, i = i + 1)* **do**
 $\quad$ $X_{i+1}, \mathcal{E}_{i+1} \leftarrow$ Build Hierarchical Tree $(X_i, \mathcal{E}_i)$ ; ▷ Calculate nodes and edges of higher level
**end**
**for** *(i = 0, i < λ, i = i + 1)* **do**
 $\quad$ $X_{i+1} \leftarrow$ Bottom-up Block $(X_i, \mathcal{E}_i)$ ;  ▷ Aggregate features for nodes of higher level
**end**
**for** *(i = λ, i > 0, i = i − 1)* **do**
 $\quad$ $X'_{i-1} \leftarrow$ Top-down Block $(X_{i-1}, X_i, \mathcal{E}_i)$ ;  ▷ Aggregate features for nodes of lower level
**end**

---

## 4 Experiments

### 4.1 Graph understanding

**Dataset Description**  To demonstrate the effectiveness of our hierarchical inductive bias, we apply Sequoia to three graph modeling datasets, namely LRGB (Dwivedi et al., 2022c), Polymer (St. John et al., 2019), and Citation network (Sen et al., 2008):

- **LRGB** (Dwivedi et al., 2022c) is a long-range graph benchmark that is built on peptides and images. In this work, we use 4 graph learning datasets: `peptides-func` (peptides), `peptides-struct` (peptides), `VOC-sp` (image), and `COCO-sp` (image). Table 1 demonstrates the statistics of these datasets. In particular, the long-range property is determined based on the graph diameter (i.e. the length of the shortest path between the most distant nodes).
- **Polymer** (St. John et al., 2019) cointains 54,000 molecules associated to polymer properties. The task in this dataset is to predict three types of density functional theory (DFT) metrics (Hohenberg & Kohn, 1964) for each polymer. In particular, the model is trained to predict the first excitation energy of the monomer (`GAP`), the energy of the highest occupied molecular orbital (`HOMO`), and the lowest unoccupied molecular orbital (`LUMO`). The predictions are evaluated and compared with the two important benchmark error levels (Faber et al., 2017): (1) *DFT errors*, the estimated average error of the DFT approximation to nature; and (2) *Chemical accuracy*, a standard target error recognized within the chemistry community.
- **Citation networks** (Sen et al., 2008) is a benchmark comprising 3 subdatasets named Cora, Citeseer, and Pubmed. In each dataset, documents are represented by sparse bag-of-words feature vectors which correspond to the nodes of the underlying graph. Besides, the edges of the graph are the citation links between documents.

**Implementation Details**  The same configuration is shared across all datasets. In particular, we set batch size = 128, hidden dimension $d = 96$, tree's layer $\Lambda = 3$, stochastic gradient descent optimizer with the initial learning rate of $10^{-4}$, maximum number of clusters $C = 32$, and epoch = 200.

Table 1: Description of datasets in the LRGB benchmark

| Dataset | # Graphs | # Nodes | # Edge | Diameter |
|---|---|---|---|---|
| Peptides-func | 15k | 2,3m | 4.7m | 56.99 ± 28.72 |
| Peptides-struct | 15k | 2,3m | 4.7m | 56.99 ± 28.72 |
| VOC-SP | 11k | 5m | 30.7m | 27.62±2.13 |
| COCO-SP | 123k | 58m | 332m | 27.39±2.14 |

Table 2: Results on **peptides property prediction**. The baseline results are taken from (Dwivedi et al., 2022b).

| Method | Params ↓ | Time ↓ | Peptides-struct | Peptides-func |
|---|---|---|---|---|
| | | | MAE ↓ | AP ↑ |
| GCN (Kipf & Welling, 2016) | 508k | 3s | 0.3496 ± 0.0013 | 0.5930 ± 0.0023 |
| GCNII (Chen et al., 2020b) | 505k | 3s | 0.3471 ± 0.0010 | 0.5543 ± 0.0078 |
| GINE (Xu et al., 2019) | 476k | 3s | 0.3547 ± 0.0045 | 0.5498 ± 0.0079 |
| GatedGCN (Bresson & Laurent, 2017) | 509k | 4.5s | 0.3420 ± 0.0013 | 0.5864 ± 0.0077 |
| GatedGCN + RWPE (Bresson & Laurent, 2017) | 506k | 4.5s | 0.3357 ± 0.0006 | 0.6069 ± 0.0035 |
| Transformer + LapPE (Vaswani et al., 2017) | 488k | 7.5s | 0.2529 ± 0.0016 | 0.6326 ± 0.0126 |
| Transformer + RWPE (Vaswani et al., 2017) | 488k | 7.5s | 0.2620 ± 0.0010 | 0.6502 ± 0.0101 |
| SAN + LapPE (Kreuzer et al., 2021) | 493k | 55s | 0.2683 ± 0.0043 | 0.6384 ± 0.0121 |
| SAN + RWPE (Kreuzer et al., 2021) | 500k | 55s | 0.2545 ± 0.0012 | 0.6562 ± 0.0075 |
| GPS (Rampášek et al., 2022) | 500k | 14s | 0.2500 ± 0.0005 | 0.6535 ± 0.0041 |
| MLP-Mixer (He et al., 2023) | 396k | 7.0s | 0.2478 ± 0.0010 | **0.6970 ± 0.0080** |
| Sequoia + RWPE (ours) | 346k | 5.3s | **0.2453 ± 0.0006** | 0.6755 ± 0.0074 |
| Sequoia + LapPE (ours) | 346k | 5.3s | 0.2526 ± 0.0019 | 0.6323 ± 0.0042 |

Table 3: Results on **VOC** and **COCO** datasets. The baseline results are taken from (Dwivedi et al., 2022b).

| Method | VOC | | | COCO | | |
|---|---|---|---|---|---|---|
| | Params ↓ | Time ↓ | macro F1 ↑ | Params ↓ | Time ↓ | macro F1 ↑ |
| GCN (Kipf & Welling, 2016) | 496k | 10.6s | 0.1268 ± 0.0060 | 509k | 116.3s | 0.0841 ± 0.0010 |
| GCNII (Chen et al., 2020b) | 492k | 8.8s | 0.1698 ± 0.0080 | 505k | 111.2s | 0.1404 ± 0.0011 |
| GINE (Xu et al., 2019) | 505k | 8.9s | 0.1265 ± 0.0076 | 515k | 105s | 0.1339 ± 0.0044 |
| GatedGCN (Bresson & Laurent, 2017) | 502k | 18.5s | 0.1265 ± 0.0076 | 509k | 189.7s | 0.2641 ± 0.0045 |
| GatedGCN + RWPE (Bresson & Laurent, 2017) | 502k | 18.5s | 0.1265 ± 0.0076 | 509k | 189.7s | 0.2574 ± 0.0034 |
| Transformer + LapPE (Vaswani et al., 2017) | 501k | 16.1s | 0.2694 ± 0.0098 | 508k | 196.1s | 0.2618 ± 0.0031 |
| Transformer + RWPE (Vaswani et al., 2017) | 501k | 16.1s | 0.2718 ± 0.0076 | 508k | 196.1s | 0.2686 ± 0.0027 |
| SAN + LapPE (Kreuzer et al., 2021) | 531k | 185s | 0.3230 ± 0.0039 | 536k | 2148s | 0.2592 ± 0.0158* |
| SAN + RWPE (Kreuzer et al., 2021) | 468k | 185s | 0.3216 ± 0.0027 | 474k | 2148s | 0.2434 ± 0.0156* |
| GPS (Rampášek et al., 2022) | 510k | 23.2s | **0.3748 ± 0.0109** | 510k | 280.5s | **0.3412 ± 0.0044** |
| Sequoia + RWPE (ours) | 438k | 13.5s | 0.3379 ± 0.0133 | 438k | 157.8s | 0.2728 ± 0.0036 |
| Sequoia + LapPE (ours) | 438k | 13.5s | 0.3481 ± 0.0116 | 438k | 157.8s | 0.2756 ± 0.0027 |

In graph-level classification, there is one layer of message passing to refine the feature before constructing the hierarchy, while in node-level segmentation, we apply four message-passing layers in this process. Thereafter we employ a combination of message-passing, hierarchical attention and residual layers once at each level of the tree to aggregate the global and local information. Additionally, we project the positional encoding (e.g., Laplacian (Dwivedi & Bresson, 2020b) or random walk (Dwivedi et al., 2022a)) to the same dimension of features ($d = 96$) and add them into the features.

**Results** Our model consistently outperforms other models in the benchmarks presented above. Regarding effective time complexity, our model is slower than message passing, yet faster than other transformer models

for graphs. More precisely, in the peptides dataset Table 2, we successfully reduce the MAE below 0.25 and surpass the other methods by a large margin, suggesting that our Cluster Learning Block constructs a useful hierarchy. We observe the same performance for the polymer dataset - Table 4. We believe that while a learnable hierarchy breaks the graph into simpler and more meaningful subgraphs, the aggregation function, with Bottom-Up and Top-Down techniques, enables efficient communication among features from different levels in the hierarchical structure. This reasoning may hold true for the polymer dataset because this data is related to chemistry, where learning from subgraphs highly benefits the network. For node-level classification, as shown in Table 3, Sequoia achieves better performances in VOC and COCO. However, the improvements are marginal for citation networks - Table 6. This aligns with the fact (Chen et al., 2020b; Dwivedi et al., 2022b) that shallow GNNs are more effective than their deep counterparts and long-range modeling methods on these benchmarks.

Table 4: Results on **polymer property prediction**. We trained all the following baselines for the Polymer datasets.

| Method | Params ↓ | Time ↓ | GAP
MAE ↓ | HOMO
MAE ↓ | LUMO
MAE ↓ |
|---|---|---|---|---|---|
| DFT error | | | 1.2 | 2.0 | 2.6 |
| Chemical accuracy | | | 0.043 | 0.043 | 0.043 |
| GCN (Kipf & Welling, 2016) | 527k | 13s | 0.1094 ± 0.0020 | 0.0648 ± 0.0005 | 0.0864 ± 0.0014 |
| GCN + Virtual Node (Kipf & Welling, 2016) | 557k | 13s | 0.0589 ± 0.0004 | 0.0458 ± 0.0007 | 0.0482 ± 0.0010 |
| GINE (Xu et al., 2019) | 527k | 11s | 0.1018 ± 0.0026 | 0.0749 ± 0.0042 | 0.0764 ± 0.0028 |
| GINE + Virtual Node (Xu et al., 2019) | 557k | 11s | 0.0870 ± 0.0040 | 0.0565 ± 0.0050 | 0.0524 ± 0.0010 |
| Transformer + LapPE (Vaswani et al., 2017) | 320k | 24s | 0.0542 ± 0.0029 | 0.0445 ± 0.0020 | 0.0458 ± 0.0057 |
| Transformer + RWPE (Vaswani et al., 2017) | 320k | 24s | 0.1280 ± 0.0122 | 0.0854 ± 0.0055 | 0.0990 ± 0.0054 |
| SAN + LapPE (Kreuzer et al., 2021) | 495k | 151s | 0.0461 ± 0.0022 | 0.0356 ± 0.0026 | 0.0356 ± 0.0017 |
| SAN + RWPE (Kreuzer et al., 2021) | 495k | 151s | 0.0467 ± 0.0031 | 0.0351 ± 0.0024 | 0.0372 ± 0.0011 |
| Sequoia + LapPE (ours) | 346k | 19s | 0.0494 ± 0.0010 | 0.0384 ± 0.0028 | 0.0376 ± 0.0035 |
| Sequoia + RWPE (ours) | 346k | 19s | **0.0409 ± 0.0009** | **0.0312 ± 0.0014** | **0.0312 ± 0.0012** |

Table 5: Sequence Classification results on Long Range Arena benchmark for efficient transformers.

| Method | IMDB
Accuracy ↑ | Cifar10
Accuracy ↑ | Listops
Accuracy ↑ | Retrieval
Accuracy ↑ | Patfhinder
Accuracy ↑ | Complexity ↓ |
|---|---|---|---|---|---|---|
| BigBird (Zaheer et al., 2020a) | 64.02 | 40.83 | 36.05 | 59.29 | 74.87 | $\mathcal{O}(n)$ |
| LongFormer (Beltagy et al., 2020a) | 62.85 | 42.22 | 35.63 | 56.89 | 69.71 | $\mathcal{O}(n)$ |
| Performer (Choromanski et al., 2021) | 65.40 | 42.77 | 18.01 | 53.82 | 77.05 | $\mathcal{O}(n)$ |
| LinFormer (Wang et al., 2020) | 53.94 | 38.56 | 35.70 | 52.27 | 76.34 | $\mathcal{O}(n)$ |
| Local Attention (Tay et al., 2020c) | 52.98 | 41.46 | 15.82 | 53.59 | 66.63 | $\mathcal{O}(n)$ |
| Linear Trans (Tay et al., 2020c) | 65.90 | 42.34 | 16.13 | 53.09 | 75.30 | $\mathcal{O}(n)$ |
| S4 (Gu et al., 2022) | 76.02 | 86.09 | **88.65** | 87.09 | *86.05* | $\mathcal{O}(n \log n)$ |
| Mega (Ma et al., 2023) | **90.43** | **90.44** | 63.14 | **91.25** | **96.01** | $\mathcal{O}(n)$ |
| Sequoia (ours) | 75.10 | 49.88 | 37.70 | 67.04 | 87.30 | $\mathcal{O}(n \log n)$ |

## 4.2 Long Sequence Modeling

**Dataset Description** We evaluate Sequoia on **Long Range Arena** (Tay et al., 2020b) (sequence classification on multiple input modalities) to demonstrate the effectiveness of our model. Our goal in the sequence modeling experiments is to validate that Sequoia is a general-purpose model and can handle multiple input modalities, including sequences of discrete tokens. Since our objective is to propose a hierarchical method that can scale to large-sized data (e.g., long sequences, large graphs, etc.) efficiently, we compare Sequoia with several efficient Transformers, including BigBird (Zaheer et al., 2020a) , LongFormer (Beltagy et al., 2020a), Performer (Choromanski et al., 2021), LinFormer (Wang et al., 2020), Local Attention (Tay et al., 2020c), and Linear Transformer (Tay et al., 2020c).

Table 6: Results on Cora, Citeseer, Pubmed

| Method | Citeseer | Cora | Pubmed | |
| --- | --- | --- | --- | --- |
| | Acc ↑ | Acc ↑ | Acc ↑ | Runtime ↓ |
| TSVM (Joachims, 1999) | 0.640 | 0.575 | 0.622 | – |
| LP(Zhu et al., 2003) | 0.453 | 0.680 | 0.630 | – |
| GRAPHEMB (Tian et al., 2014) | 0.432 | 0.672 | 0.653 | – |
| PLANETOID-G (Yang et al., 2016) | 0.493 | 0.691 | 0.664 | – |
| PLANETOID-T (Yang et al., 2016) | 0.629 | 0.757 | 0.757 | – |
| ChebNet (Defferrard et al., 2016) | 0.698 | 0.780 | 0.744 | – |
| GCN (Kipf & Welling, 2016) | 0.703 | 0.815 | 0.790 | 0.32s |
| GAT (Veličković et al., 2018) | 0.725 | 0.830 | 0.790 | 2.42s |
| Sequoia (ours) | **0.732** | **0.845** | **0.794** | 2.15s |

**Implementation Details**   For experimental simplicity, we use a standard model backbone configuration throughout all sequence classification tasks. More specifically, our model consists of four transformer layers, with a hidden dimension of 512 splits across settings for all datasets. We use a dropout rate of 0.1, and our model is optimized by Adam optimizer with a learning rate of 0.001 with a linear warmup. The accuracy reported corresponds to the test accuracy obtained on the best-performing model according to validation steps over every 2,000 training steps.

**Results**   Table 5 reports our results for long-range sequence modelling and by the introduction of hierarchical interactions, Sequoia surpasses other conventional efficient transformers on LRA. One possible reason for the improvements is that during forming a hierarchy, we explicitly divide sentences or images into smaller chunks which facilitates aggregation process of the model. Otherwise, other approaches in the table let the network process entire sentences or images.

### 4.3  Point clouds

**Dataset Description**   In order to demonstrate the efficiency of Sequoia on point clouds, we conduct experiments on two tasks:

- **Shape classification:** ModelNet40 (Wu et al., 2015) dataset is a classification dataset that contains 12,311 3D models categorised into 40 classes.

- **Part segmentation:** ShapeNetPart (Mo et al., 2019) dataset is a part segmentation dataset, which contains 16,881 synthetic point clouds from 16 classes.

**Implementation Details**   In all experiments, our model use four Transformer layers and 4-layer tree. The other configurations are kept the same as other works. In particular, the optimizer used in both experiments is SGD (learning rate = 0.05, momentum = 0.9, and weight decay = 0.0001) and we train the model for 200 epochs with a batch size of 32 for classification and 16 for segmentation.

**Results**   As observed from Table 7 and 8, Sequoia achieves competitive results on point cloud classification, but it does not seem to perform as well as other baselines in part segmentation. The reason is that apart from PointNet (Qi et al., 2017a), the remaining models in this experiment already use hierarchical structure in their design, so our method obtains no significant superior in terms of aggregating features, compared to them. However, with the new algorithm for tree construction, we can avoid $O(n^2)$ complexity when calculating distance for sampling and finding nearest neighbors in other point-based methods.

Table 7: Shape Classification results on ModelNet40

| Method | Accuracy ↑ | mAcc ↑ | Params ↓ | Time ↓ |
|--------|-----------|--------|----------|--------|
| VoxNet (Maturana & Scherer, 2015) | 85.9 | 83.0 | – | – |
| MVCNN (Su et al., 2015) | 90.1 | – | – | – |
| PointNet (Qi et al., 2017a) | 89.2 | 86.0 | 0.6M | 0.81ms |
| DGCNN (Wang et al., 2019) | 92.9 | 90.2 | 1.82M | – |
| GridGCN (Xu et al., 2020) | 93.1 | **91.3** | – | 42.2ms |
| Set Transformer (Lee et al., 2019a) | 90.4 | – | – | – |
| PointNet++ (Qi et al., 2017b) | 91.9 | 88.4 | 1.0M | 1.98ms |
| PointConv (Wu et al., 2019) | 92.5 | – | – | – |
| KPConv (Thomas et al., 2019) | 92.9 | – | 14.3M | 543.7ms |
| PointTransformer (Zhao et al., 2021) | **93.7** | 90.6 | 11.7M | 2.0ms |
| Sequoia (ours) | 92.0 | 88.4 | 10.9M | 9.6ms |

Table 8: Part Segmentation results on ShapeNetPart

| Method | Inst IoU ↑ | Class IoU ↑ | Params ↓ | Time ↓ |
|--------|-----------|-------------|----------|--------|
| PointNet (Qi et al., 2017a) | 71.9 | 43.7 | 3.6M | 1.0 ms |
| DGCNN (Wang et al., 2019) | 85.1 | 82.3 | 1.3M | 125ms |
| PointNet++ (Qi et al., 2017b) | 85.1 | 81.9 | 1.0M | 1.41ms |
| PointConv (Wu et al., 2019) | 85.7 | 82.6 | – | – |
| PointCNN (Li et al., 2018) | 86.1 | **84.6** | – | – |
| PointTransformer (Zhao et al., 2021) | **86.6** | 83.7 | 7.8M | 2.69ms |
| Sequoia (ours) | 83.8 | 80.6 | 7.26M | 7.8ms |

## 4.4 Ablation study

To comprehensively evaluate our model, we conduct ablation studies on several aspects including graph sizes, the maximum number of clusters, different types of hierarchy, the number of layers, and different types of attention blocks.

Table 9: Ablation study for complexity and memory.

| Method | Nodes | Synthetic Graph | | |
|--------|-------|-----------------|---|---|
| | | Time ↓ | Memory ↓ | Parameters ↓ |
| Message Passing | | 0.0156s | 1.171GB | 350k |
| Attention | 100 | 0.0234s | 1.175GB | 500k |
| Sequoia (ours) | | 0.0221s | 1.187GB | 437k |
| Message Passing | | 0.0237s | 1.447GB | 350k |
| Attention | 1000 | 0.0379s | 1.635GB | 500k |
| Sequoia (ours) | | 0.0235s | 1.355GB | 437k |
| Message Passing | | 0.5152 | 20.295GB | 350k |
| Attention | 10000 | OOM | OOM | 500k |
| Sequoia (ours) | | 0.5081 | 19.563GB | 437k |

**Scalability** We generate synthetic graphs with varying sizes from 100 to 10,000 nodes and perform node classification. Table 9 shows that our model is comparable to message-passing networks in terms of efficiency.

**Different types of hierarchy** We compare 3 types of hierarchy in Table 10

Table 10: Ablation study on clusters and hierarchy.

| Cluster | Peptides-func | VOC | Hierarchical | Peptides-func | VOC |
| --- | --- | --- | --- | --- | --- |
| | Accuracy ↑ | macro F1 ↑ | | Accuracy ↑ | macro F1 ↑ |
| 16 | 66.95 | 0.3236 | Fixed | 62.78 | 31.83 |
| 32 | **67.55** | **0.3481** | Random | 66.19 | 31.20 |
| 64 | 65.67 | 0.3435 | Learnable | **67.55** | **34.81** |

Table 11: Ablation study on tree's layers and attention blocks.

| # Layers | ModelNet40 | Attention Blocks | ModelNet40 |
| --- | --- | --- | --- |
| | Accuracy ↑ | | Accuracy ↑ |
| 3 | 90.19 | Children | 87.50 |
| 4 | **92.00** | Children + Siblings | 91.50 |
| 5 | 89.87 | Children + Siblings + Ancestors | **92.00** |

Table 12: Ablation study on hierarchical structure

| Method | Inst IoU | Class IoU |
| --- | --- | --- |
| Non-hierarchy | 78.8 | 74.9 |
| Sequoia's hierarchy | **82.7** | **79.5** |

- In a **fixed hierarchical structure**, we initially assign nodes to a predetermined hierarchical framework, where clusters are defined based on the type of the dataset (e.g. KNN for point clouds, k ring for graphs, sentence chunking for texts, and sliding window for images).
- In a **random hierarchical structure**, we assign nodes to a randomly generated hierarchical framework.
- In our **learnable hierarchical structure**, the model learns the optimized hierarchical structure.

By constructing the meaningful hierarchical structure, our methods achieves best performance, compared to the others.

**Other configurations**   We run our model on several settings in terms of the maximum number of clusters, the number of layers, and different types of attention blocks. The results are reported in Table 10, 11, and 12.

### 4.5   Other implementation details

**Fairness in comparison between our method and the baselines**   For Tables 2, 3, 4, we reran all the models in the benchmark using the same framework GraphGPS (Rampášek et al., 2022) on same devices (RTX3090) for fair comparison in terms of runtime and number of parameters. We use (Fey & Lenssen, 2019) to evaluate the run-time for Table 6. There is no unified framework for point clouds, so we mainly obtained information based on the benchmarks of other papers such as PointNext(Qian et al., 2022) and RepSurf(Ran et al., 2022) for fair comparison in Table 7 and 8. As for Table 5, since our code does not use the same framework as other efficient transformers, we to report the Big-O complexity instead. In terms of the training time, we followed the same configuration when training and all our experiments had the same number of epochs as the baselines, but the time of each epoch was different.

**Tuning the maximum number of clusters**   In the Adaptive Hierarchy, the number of cluster $k$ is predefined meaningfully based on the statistics computed from the dataset. For example, $k$ may be the number of functional groups existing in a molecule, the number of parts of a 3D objects, or the number of

elements inside a sentence. Furthermore, it is unnecessary to define $k$ precisely since our Adaptive Hierarchy block allows for empty clusters, which means we only need to approximate the upper bound of the number of actual clusters. Additionally, the challenge of defining $k$ could be eased by the multi-level mechanism of the structure. Specifically, when the number of the predefined clusters is too small to capture the entire functional sub-graphs, they can be extracted at a higher level in the tree. In short, during the experiments, we merely select a reasonable upper-bound $k$ that fits does not cause GPU memory issues rather than putting effort into tailoring the best structure by tuning $k$ for each level in the hierarchy.

**The flexibility of Sequoia's Adaptive Hierarchy** Indeed, our Adaptive Hierarchy is a standalone block that can be potentially viewed as a plug-and-play module for other graph transformer models. Theoretically, the hierarchical structure can potentially benefit other methods that require a hierarchical organization of the data. In our paper, our primary proposal is hierarchical "attention", aimed at maximizing interactions between different levels in the hierarchy generated by the Adaptive Hierarchy block, which can be easily integrated to other architectures.

## Conclusion

The first ingredient of our model, self-attention, has proved expressive and invaluable in learning powerful models on a variety of structured data. The second ingredient, hierarchy, provides a useful inductive bias and provides a convenient structure for both the organization of representations and the optimization of computational budget. This paper introduces a mechanism for merging these in a single **geometric learning building block**. We found this combination to be particularly useful for graphs that involve long-range interactions. Our ablation studies suggest that in terms of computation, it compares with message-passing methods, even **surpassing them on larger graphs**.

## Acknowledgements

SR and HS want to acknowledge funding support from FPT, CIFAR, and NSERC. Computational resources were in part provided by Mila and Compute Canada.

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

## A  Transformers as general message-passing networks

**Transformer block**   Let $L$ be the number of layers, $H$ the number of attention heads, $N$ the maximum sequence length, and $d$ the token feature dimension at the output of the transformer block [1]. A Transformer is defined by the iterative composition of *transformer blocks* sharing the following form: $\mathcal{T}(x) = \mathcal{F}(\mathcal{A}(x)+x)$.

**Scaled dot-product self-attention**   Scaled-dot-product attention updates embeddings through a weighted pooling of token-wise features pooled across the entire input sequence. Let us first compute (`query`, `key`, `value`) pairs for each token:

$$Q = XW_Q, \quad K = XW_K, \quad V = XW_V,$$

with $W_Q \in \mathbb{R}^{d \times d_k}$, $W_K \in \mathbb{R}^{d \times d_k}$ and $W_V \in \mathbb{R}^{d \times d_v}$. Token $i$-th's embedding is updated using the following formula:

$$X'_i = \mathcal{A}_{W_Q, W_K, W_V}(X, i) = \sum_{j \in [1, N]} \mathcal{S}(Q_i, K_j) V_j.$$

In scaled-dot-product attention, the interaction scores $\mathcal{S}(\cdot, \cdot)$ are computed as follows:

$$\mathcal{S}(Q_i, K_j) = \frac{\exp(Q_i K_j^T)}{|\mathcal{S}_i|} \times \frac{1}{\sqrt{d_k}},$$

with $|\mathcal{S}_i|$ the normalization factor associated with token $i$:

$$|\mathcal{S}_i| = \sum_{j \in [1, N]} \exp(Q_i K_j^T).$$

**Multiheaded attention**   Multiheaded *self-attention* parallelizes the computation of attention into various subspaces of the latent space. We first split each input into $H$ different subspaces of latent dimension $d_k^h = \frac{d_k}{H}$ for queries and keys and $d_v^h = \frac{d_v}{H}$ :

$$\mathcal{A}_{\text{multiheaded}}(X) = [X'^1 :: \cdots :: X'^h :: \cdots :: X'^H] W_O,$$

where $X'^h$ is the output of the self-attention mechanism for head $h \in [1, H]$:

$$\mathcal{A}_{\text{singlehead}}(X) = \mathcal{A}_{W_Q^h, W_K^h, W_V^h}(X),$$

with $W_Q^h \in \mathbb{R}^{d \times d_k^h}, W_K^h \in \mathbb{R}^{d \times d_k^h}$, $W_V^h \in \mathbb{R}^{d \times d_v^h}$ and $W_O \in \mathbb{R}^{d_v \times d}$.

**Message passing neural networks**   MPNNs iteratively apply a **local message passing equation** updating a node's hidden state $m_i^{t+1}$ based on its *neighbours'* embeddings. This summarized representation of its local context is then fused with its previous embedding $X_i^t$ in order to compute its updated embedding $X_i^t$ [2]. In the limit of infinitely many layers, soft attention can be understood as a simple message-passing mechanism deprived of the locality inductive bias (Kim et al., 2022b) (the aggregating neighbourhood of token $i$ is the entire sequence/token set itself $\mathcal{N}(i) = [0, n-1]$).

$$
\begin{aligned}
m_i^{t+1} &= \Xi\left(\{\chi(X_i^t, X_j^t, e_{i \to j})\}_{j \in \mathcal{N}(i)}\right) \\
X_i^{t+1} &= \zeta\left(m_i^{t+1}, X_i^t\right)
\end{aligned}
\tag{3}
$$

*Most efficient transformer models employing locality-sensitive heuristics can be adequately framed under the simple message-passing framework outlined above.*

---

[1] For the sake of simplicity, we omit the layer and head indices $l$ and $h$, the extension to the multi-layer / multi-head case being straightforward.

[2] For that purpose, a gated mechanism is often employed in order to adaptively filter past information based on its relevance for the task at hand

**Transformers for point clouds** The hierarchical structure has existed in point cloud models since Point-Net++ (Qi et al., 2017b), which significantly improved upon its non-hierarchical version predecessor PointNet (Qi et al., 2017a). However, the attention operation was only successfully integrated into this dataset in Point Cloud Transformer (Guo et al., 2021). This work applied self-attention to all points in the point cloud and gained outstanding results on object classification. Afterward, Point Transformer (Zhao et al., 2021) leveraged the hierarchy with local attention to enhance the performance on the segmentation problems on the point cloud. Following that, Stratified Point Transformer (Lai et al., 2022) adapted voxels in order to estimate local regions for attention and positional encoding, further boosting the performance of the Transformer model in point cloud semantic segmentation. Meanwhile, another paper named Fast Point Transformer (Park et al., 2022) tries to optimize the attention function and proposes a new attention-based model that is 129 times faster than Point Transformer (Zhao et al., 2021) in semantic segmentation. In our work, we not only apply a new hierarchical scheme built through voxelization but also propose a new way to aggregate information across the levels of the hierarchy.

## B    Proofs of Computational Complexity

The total number of nodes (both true and virtual) in the tree is :

$$|I| = \sum_{\lambda=0}^{\Lambda-1} |I_\lambda| = \sum_{\lambda=0}^{\Lambda-1} \lceil n \times \frac{1}{k^\lambda} \rceil = n \times \left[ \frac{1 - \left(\frac{1}{k}\right)^\Lambda}{1 - \frac{1}{k}} \right] + \mathcal{O}(\Lambda)$$

$$= n \left[ \frac{k^\Lambda - 1}{k^{\Lambda-1}(k-1)} \right] + \mathcal{O}(\Lambda)$$

$$\approx (\leq) \, n \times \underbrace{\left\lfloor \frac{k}{k-1} \right\rfloor}_{\Delta_k}$$

Thus, $|I| \leq n \times \Delta_k$, with $\Delta_k = \frac{k}{k-1} \geq 1$ being the **dilation factor** of the tree. Let us now examine the computational complexity of `Sequoia`'s efficient attention scheme, both in terms of *runtime* and *memory* :

$$\mathcal{C}_{\texttt{Sequoia}}(n) = \mathcal{C}_{\text{children}}(n) + \mathcal{C}_{\text{siblings}}(n) + \mathcal{C}_{\text{ancestors}}(n)$$

$$\mathcal{C}_{\text{children}}(n) = \sum_{\lambda=1}^{\Lambda-1} \mathcal{C}_{\text{children}}(n, \lambda),$$

$$\mathcal{C}_{\text{siblings}}(n) = \sum_{\lambda=0}^{\Lambda-1} \mathcal{C}_{\text{siblings}}(n, \lambda),$$

$$\mathcal{C}_{\text{ancestors}}(n) = \sum_{\lambda=0}^{\Lambda-2} \mathcal{C}_{\text{ancestors}}(n, \lambda)$$

$$\forall \lambda \in [0, \Lambda-1] \, \mathcal{C}_{\text{children}}(n, \lambda) = \mathcal{O}(|I_\lambda| \times k)$$
$$\forall \lambda \in [0, \Lambda-1] \, \mathcal{C}_{\text{siblings}}(n, \lambda) = \mathcal{O}(|I_\lambda| \times k)$$
$$\forall \lambda \in [0, \Lambda-1] \, \mathcal{C}_{\text{ancestors}}(n, \lambda) = \mathcal{O}(|I_\lambda| \times (\Lambda - \lambda))$$
$$= \mathcal{O}(|I_\lambda| \times \Lambda)$$

Using the upper bound on the number of nodes in the tree constructed by `Sequoia`, we get :

$$\mathcal{C}_{\text{children}}(n) = \mathcal{O}\left(\sum_{\lambda=0}^{\Lambda-1} |I_\lambda| \times k\right) \approx (\leq) \, n \times k \times \Delta_k$$

$$\mathcal{C}_{\text{ancestors}}(n) = \mathcal{O}\left(\sum_{\lambda=0}^{\Lambda-1} |I_\lambda| \times \Lambda\right) \approx (\leq) \, n \times \Lambda \times \Delta_k$$

For $\mathcal{C}_{\text{siblings}}(n)$, we obtain a similar bound as for $\mathcal{C}_{\text{children}}(n)$. We finally derive the following class of *quasi-linear* computational complexity for our proposed attention model :

$$\mathcal{C}_{\texttt{Sequoia}}(n) = \mathcal{O}\left(n \times \left[2 \times \left[\frac{k}{k-1}\right] + \Lambda\right]\right)$$
$$= \mathcal{O}(n \times (k + \log(n))$$

Indeed, the tree depth is such that $\Lambda = \lceil \frac{\log(n)}{\log(k)} \rceil \leq \frac{\log(n)}{\log(k)} + 1 = \mathcal{O}(\log(n))$. In particular, we notice that this upper bound on $\mathcal{C}_{\texttt{Sequoia}}(n)$ does not depend on the mode of computation of attention inside a given transformer block (`bottom-up` / `top-down` or `parallel`), and that the total complexity of `Sequoia`'s attention scheme is dominated by the attention computed at the first level of the tree[3].

## C  Visualization

In this section, we extract the feature of nodes in different levels of the tree including leaf nodes, nodes from the low-level, nodes from the high-level, and root nodes. Then, we use t-SNE (van der Maaten & Hinton, 2008) to reduce the features' dimension into 2 and visualize these features in Figures 4, 5, 6, and 7. We observe a clear clustering pattern from these figures that suggests our hierarchical method has learned useful and meaningful representation.

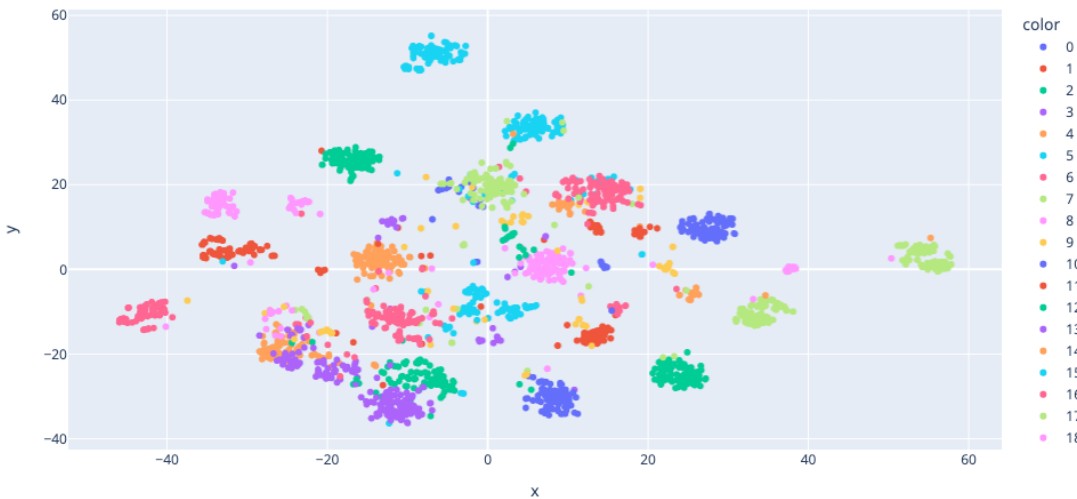

Figure 4:  Visualization of the feature of root nodes in Shape Classification task.

---

[3]Precisely, *siblings* attention on the first level and, to a lesser extent, ancestors attention on the first level.

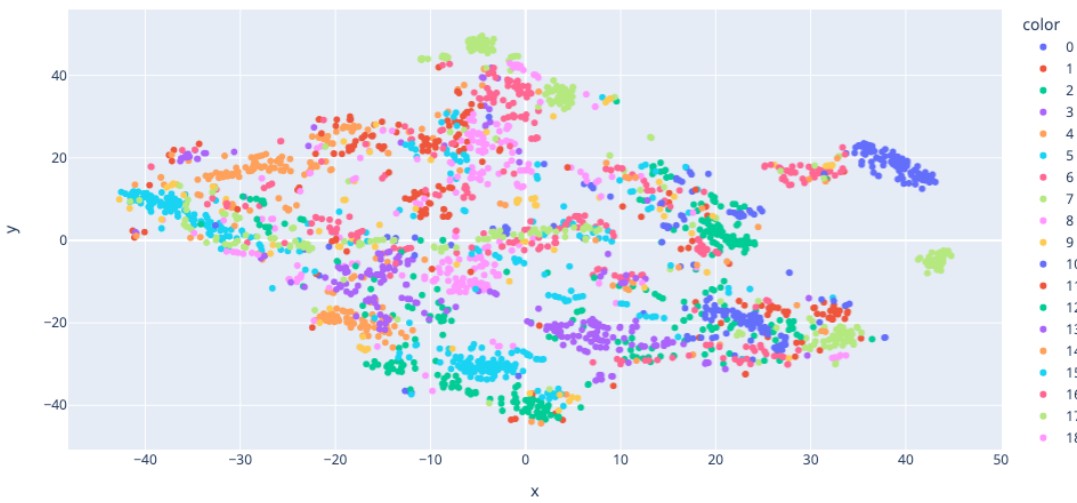

Figure 5: Visualization of the feature of nodes on the high level of the tree in Shape Classification task.

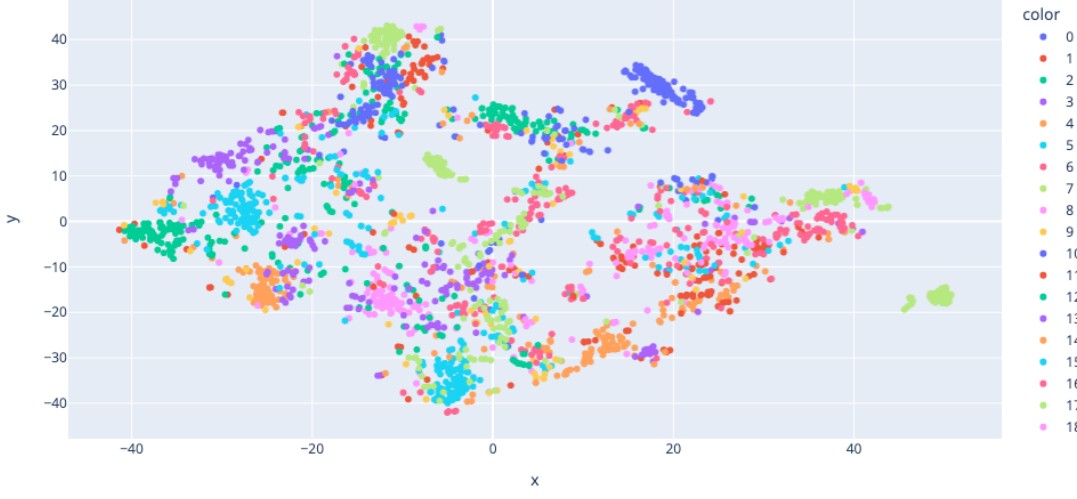

Figure 6: Visualization of the feature of nodes on the low level of the tree in Shape Classification task.

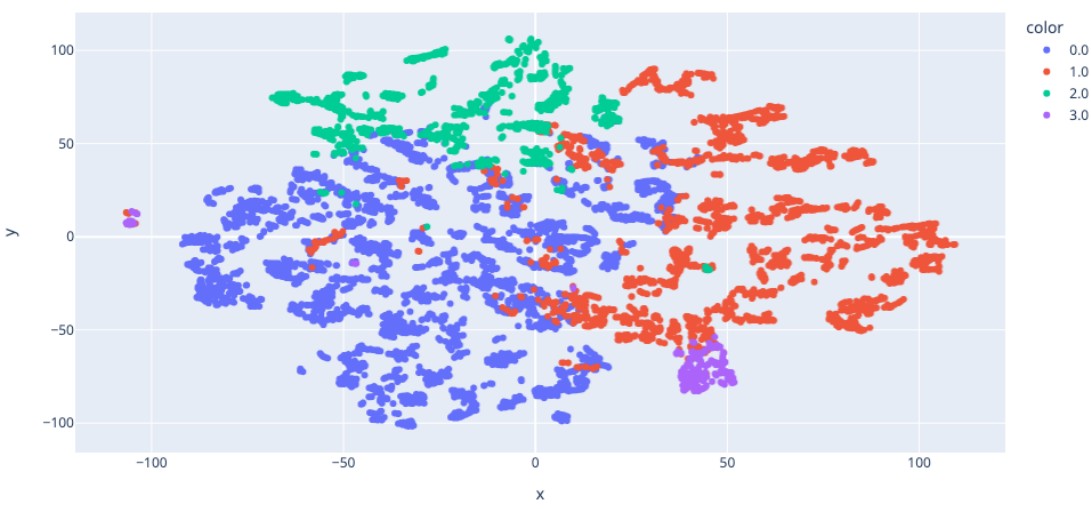

Figure 7: Visualization of the feature of leave nodes on Part Segmentation task.

