# OpenReview forum: "Scalable Hierarchical Self-Attention with Learnable Hierarchy for Long-Range Interactions"
_TMLR — Accepted by TMLR_

### Review · Reviewer_u9PB · 2024-01-26

**Summary Of Contributions:**

The authors propose an extension of the transformer architecture to graphs, where multiple layers of attention are performed within different levels of clustering of the nodes of the graph. This enables information to be pooled at various resolutions while avoiding the quadratic complexity of a standard transformer. The clustering is done by a message-passing neural network that is trained end-to-end for the task.

**Audience:**

Yes

**Claims And Evidence:**

Yes

**Requested Changes:**

My first reaction when looking at the results is to be quite skeptical. How can you have so many significant digits in your analysis? How is this statistically justifiable? Also, where do the error bars come from? Similarly, how are the reported timing numbers fair? Was everything implemented in a common software framework, run on the same hardware, and given a similar level of attention to optimizing the code?

The paper would benefit from a formal problem definition. For example, do nodes have features but not edges? How would your approach be updated to accommodate edge features? Relatedly, I was confused in sec 3. where things were described in terms of tokens instead of nodes+edges.

big-O analysis of the compute requirements for your modeling approach are insufficient in the modern where everything is vectorized. For example, many mpnn implementations assume a fully-connected graph, since this is easier to vectorize than message passing on a sparsely-connected graph. Your claims about nlogn complexity are inadequate as it does mention the practicality of achieving this complexity on modern hardware. Can you discuss the practicality, given that the structure of the graph is dynamic?

The concept of 'input size'  was poorly defined. Is the idea that your method has complexity that is linear in the number of edges? The stated complexity is nlogn; is n the number of edges? Are you assuming that the number of edges is << num_nodes ** 2? Otherwise, the mpnn stage is quadratic in the number of nodes, just like a normal transformer.

**Strengths And Weaknesses:**

**Strengths
Strong results across a variety of popular benchmarks. Efficient transformers are a hot topic that many readers will be interested in.

Related work is presented well, and the technical contribution of the paper is well motivated.

**Weaknesses

My 'requested changes' below raises a number of key questions/issues I have.

---

> ### Author Response · Authors · 2024-02-04
> **Rebuttal for Reviewer u9PB**
>
> We would like to thank the reviewer for their useful feedback on the information that may unclear in the paper. We will provide the answers for all of the questions below and updated the paper based on this feedback.
>
> **How can you have so many significant digits in your analysis? How is this statistically justifiable? Also, where do the error bars come from?**
>
> We follow the format used in other previous works on this benchmark, where results are reported with four decimal digits. Note that the statistical significance is quantified by the reported error bars that are standard deviation using five random seeds. We provide a table below to show the difference in terms of performance, compared to previous SOTA.
>
> | **Method** | **Peptides-struct** MAE ↓ |  **Peptides-func** AP ↑  |
> |:-----------------------|:------------:|:----------:|
> | GCN [3]                 |      0.3496 ± 0.0013      |     0.5930 ± 0.0023      |
> | GCNII [4]                 |      0.3471 ± 0.0010      |     0.5543 ± 0.0078      |
> | GINE [5]                  |      0.3547 ± 0.0045      |     0.5498 ± 0.0079      |
> | GatedGCN [6]              |     0.3420 ± 0.0013      |     0.5864 ± 0.0077      |
> | GatedGCN + RWPE [6]        |      0.3357 ± 0.0006      |     0.6069 ± 0.0035      |
> | Transformer + LapPE   [7] |      0.2529 ± 0.0016      |     0.6326 ± 0.0126      |
> | Transformer + RWPE   [7]  |      0.2620 ± 0.0010      |     0.6502 ± 0.0101      |
> | SAN + LapPE        [8]    |      0.2683 ± 0.0043      |     0.6384 ± 0.0121      |
> | SAN + RWPE         [8]    |     0.2545 ± 0.0012      |     0.6562 ± 0.0075      |
> | GPS              [9]      |      0.2500 ± 0.0005      |     0.6535 ± 0.0041      |
> | MLP-Mixer          [10]    |  0.2478 ± 0.0010 (0.00%)  | 0.6970 ± 0.0080 (0.00%)  |
> ||||
> | Sequoia + RWPE (ours)  | 0.2453 ± 0.0006 (+0.82%)  | 0.6755 ± 0.0074 (-3.16%) |
> | Sequoia + LapPE (ours) | 0.2526 ± 0.0019 (-1.98%)  | 0.6323 ± 0.0042 (-9.33%) |
>
> **Similarly, how are the reported timing numbers fair? Was everything implemented in a common software framework, run on the same hardware, and given a similar level of attention to optimizing the code?**
>
> We can ensure that the time comparison is fair because we use the same hardware for all the runs, which is the GPU 3090, when measuring the time complexity. Additionally, our code is based on the previously published code, ensuring consistency in attention to optimizing the code. In short, all the experiments in the table use the same framework (Graph GPS[1]) and libraries (PyTorch Geometric and PyTorch).
>
> **For example, do nodes have features but not edges?**
>
> We only use node features when applying hierarchical attention, as the edges feature is already encoded into nodes in previous steps including positional encoding process and message passing before doing attention, where we aggregate the features for nodes using information related to edges. Therefore, in section 3, we consider those node features as tokens. In addition, we keep the edge feature fixed after forming a hierarchy, and only update node features in later steps.
>
> **References:**
>
> [1] Ladislav Rampášek et al. Recipe for a General, Powerful, Scalable Graph Transformer. Advances in Neural Information Processing Systems, 35, 2022.
>
> [2] Vijay Prakash Dwivedi et al. Long range graph benchmark. arXiv:2206.08164, 2022.
>
> [3] Thomas N Kipf and Max Welling. Semi-supervised classification with graph convolutional networks. arXiv preprint arXiv:1609.02907, 2016
>
> [4] Ming Chen et al. Simple and deep graph convolutional networks. In Proceedings of the 37th International Conference
> on Machine Learning, volume 119 of Proceedings of Machine Learning Research, pp. 1725–1735. PMLR, 13–18 Jul 2020.
>
> [5] Keyulu Xu et al. How powerful are graph neural networks? In 7th International Conference on Learning Representations, ICLR 2019, New Orleans, LA, USA, May 6-9, 2019.
>
> [6] Xavier Bresson and Thomas Laurent. Residual gated graph convnets. CoRR, abs/1711.07553, 2017. URL http://arxiv.org/abs/1711.07553
>
> [7] Ashish Vaswani et al. Attention is all you need. Advances in neural information processing systems, 30, 2017.
>
> [8] Devin Kreuzer et al. Rethinking graph transformers with spectral attention. In Advances in Neural Information Processing Systems, volume 34, pp. 21618–21629. Curran Associates, Inc., 2021.
>
> [9] Ladislav Rampášek et al. Recipe for a General, Powerful, Scalable Graph Transformer. Advances in Neural Information
> Processing Systems, 35, 2022
>
> [10] Xiaoxin He et al. A generalization of ViT/MLP-mixer to graphs. In Proceedings of the 40th International Conference on Machine
> Learning, volume 202 of Proceedings of Machine Learning Research, pp. 12724–12745. PMLR, 2023.

---

> ### Author Response · Authors · 2024-02-04
> **Rebuttal for Reviewer u9PB (continued)**
>
> **Big-O analysis of the compute requirements for your modeling approach are insufficient in the modern where everything is vectorized. For example, many mpnn implementations assume a fully-connected graph, since this is easier to vectorize than message passing on a sparsely-connected graph. Your claims about nlogn complexity are inadequate as it does mention the practicality of achieving this complexity on modern hardware. Can you discuss the practicality, given that the structure of the graph is dynamic?**
>
> We agree with the reviewer, and that is why we also report the practical run-time. We would like to note that we parallelized the updates as seen in Eqs 1 and 2 in Section 3.1. Moreover, in our implementation the attention mechanism is performed in parallel for all nodes within a single resolution. Finally, the use of PyTorch Geometric optimizes MPNN implementations for sparsely-connected graphs.
>
> **The concept of ’input size’ was poorly defined. Is the idea that your method has complexity that is linear in the number of edges?**
>
> We consider the input size as the number of nodes rather than the number of edges.
>
> **The stated complexity is nlogn; is *n* the number of edges? Are you assuming that the number of edges is much less than num nodes square? Otherwise, the mpnn stage is quadratic in the number of nodes, just like a normal transformer.**
>
> In our graph datasets, we have experimented with long-range graphs (that have long diameters, i.e. the maximum shortest path) from the Long-Range Graph Benchmark (LRGB)[2].
>
> * For the peptides / proteins dataset of LRGB, the average number of nodes is 150.94, and the average number of edges is 307.30 with the average edge density (i.e. number of edges divided by the maximum number of edges that a graph can have) is 0.027.
> * For the two computer vision graph datasets of LRGB, the average numbers of nodes are 476.88 and 479.40, and the average numbers of edges are 2,693.67 to 2,710.48, respectively. The average edge density is 0.024.
>
> Fundamentally, these long-range graphs are sparse, i.e. the number of edges is a constant times the number of nodes with the constant is roughly between 2 and 5. We can say that |E| = O(|V|) for these graphs.
>
> Our hierarchical transformer is efficient, and has the time and space complexities in terms of the number of nodes, that much smaller / better than the quadratic complexities of the conventional transformer.
>
> The MPNN procedure used in our implementation is based on propagation along the edges and has time complexicity proportional to the number of edges, and does not have quadratic complexities in both time and space.
>
> **References:**
>
> [2] Vijay Prakash Dwivedi, Ladislav Rampášek, Mikhail Galkin, Ali Parviz, Guy Wolf, Anh Tuan Luu, and Dominique Beaini. Long range graph benchmark. arXiv:2206.08164, 2022.

---

> ### Comment · Reviewer_u9PB · 2024-03-12
> **Please respond to my review**
>
> The reviewers need to achieve consensus about this paper. However, I raised a number of questions that have not been addressed in a response. Can you please respond to my review?

---

> ### Author Response · Authors · 2024-03-12
> **Thank you for your help!**
>
> Dear Reviewer u9PB,
>
> Thank you so much for raising the questions! We sincerely apologize for our mistake. We have answered all your questions in our rebuttal before, but the rebuttal was invisible to you (and only visible to the editor). We fixed the visibility issue (by setting the visibility to Everyone) and hope that you will consider our answers. Please let us know if you can see our rebuttal and whether you need any further response from us! Thank you again!
>
> Best regards,
> Authors

---

> > ### Comment · Reviewer_u9PB · 2024-03-12
> > **Thanks!**
> >
> > I'm glad we cleared up the issue. Thank you so much for the detailed response.

---

### Review · Reviewer_y1CR · 2024-02-07

**Summary Of Contributions:**

The authors propose a new architecture for efficient self-attention by learning a hierarchy with a message passing network and Gumbel-softmax-based clustering at each level of the architecture. The claimed benefits by the authors are coming from the fact that the Gumbel-softmax allows for reducing inference latency, the incorporation of bottom-up, top a-down and intra attention contributions, as well as the log-complexity arising from the k-ary tree structure.

**Audience:**

Yes

**Claims And Evidence:**

Yes

**Requested Changes:**

The second paragraph of the introduction misses citations. When you say that your "initial motivation coincides with a similar idea" you should cite what you mean.

Please use \citep as suggested by the citation style. For example, when citing Reformer, you should you \citep instead of \citet.

I did not understand claim 1 in section 2. What do you mean that these methods cannot be used on other modalities? Can you give a more concrete example to understand it?

Can you give more details (maybe in the caption) about the latency benchmarking in Table 2? Question also extends to the rest of the tables.

Tables 5-7 are missing any parameter count and latency measurements. Why? Please add them where you can.

You should also include SOTA (for reference) on the Long Range Arena in Table 5. Likewise for Table 6.

In Table 10, why does a random hierarchy for 32 clusters perform the best?

**Strengths And Weaknesses:**

Strengths:

1. The authors propose a sensible architecture, Sequoia, for learning hierarchical self-attention.

2. The experiments across many different tasks and datasets signify the usefulness of Sequoia.

Weaknesses:

1. The paper is missing interpretations on concrete examples. The visualizations in the appendix are not very informative because they don't delve into a connection between the hierarchies that your model learns, and the semantics of the dataset.

2. I believe the benchmarking time for training is omitted from discussion in the paper, but the authors should discuss how it compares with the baselines.

3. The contributions from each type of attention (described in Section 3.2) are not ablated.

---

> ### Author Response · Authors · 2024-02-21
> **Rebuttal for Reviewer y1CR**
>
> First of all, we would like to thank the reviewer for the constructive feedback. We detail our response as follows.
>
> **Weaknesses:**
>
> - **The paper is missing interpretations of concrete examples. The visualizations in the appendix are not very informative because they don't delve into a connection between the hierarchies that your model learns, and the semantics of the dataset.**
>
>     As illustrated in Figures 4, 5, 6, and 7 in Appendix C, our proposed framework can learn meaningful low-dimensional embeddings of point clouds at different level of the hierarchy (i.e. root, leave, or intermediate nodes). The visualizations demonstrate that the embeddings at different levels are separated into multiple clusters corresponding to the classes of the entire shapes. This indicates the effective communication between the nodes along the hierarchy via attention mechanisms.
>
> - **I believe the benchmarking time for training is omitted from the discussion in the paper, but the authors should discuss how it compares with the baselines.**
>
>     In terms of the training time, we followed the same configuration when training and all our experiments had the same number of epochs as the baselines, but the time of each epoch was different.
>
> - **The contributions from each type of attention (described in Section 3.2) are not ablated.**
>
>     We have included an ablation study regarding the contributions from each type of attention in Table 11 of Section 4.4.
>
> **Requested Changes:**
>
> - **The second paragraph of the introduction misses citations. When you say that your "initial motivation coincides with a similar idea" you should cite what you mean.**
>
>     This sentence links our work with the others that aim to reduce the quadratic-complexity computations of vanilla transformers to linear or log-linear complexities such as Performer, Linear Transformer, etc. We have already cited these works in our manuscript.
>
> - **Please use \citep as suggested by the citation style. For example, when citing Reformer, you should you \citep instead of \citet.**
>
>     Yes, we have changed those citations in our manuscript and will upload the revision when given a chance.
>
> - **I did not understand claim 1 in section 2. What do you mean that these methods cannot be used on other modalities? Can you give a more concrete example to understand it?**
>
>     In our manuscript, we have mentioned several works that employ efficient Transformers for learning on texts, such as Performer or Linear Transformer. While these works perform well on sequences, they lack important inductive biases on graphs, such as different sizes of neighborhoods, and unordered structures. It is important to note that our architecture can work and achieve competitive results with multiple types of data including point clouds, long-range sequences, and long-range graphs.

---

> ### Author Response · Authors · 2024-02-21
> **Rebuttal for Reviewer y1CR (continued)**
>
> - **Can you give more details (maybe in the caption) about the latency benchmarking in Table 2? Question also extends to the rest of the tables.**
>
>     Regarding Table 2-4, we reran all the models in the benchmark using the same framework, GraphGPS, for fair comparison. However, there is no unified framework for point clouds, so we mainly obtained information based on the benchmarks of other papers such as PointNext and RepSurf for fair comparison. We also relied on PyTorch Geometric to measure the time for Table 6. As for Table 5, since our code does not use the same framework as other efficient transformers, we would like to report the Big-O complexity instead.
>
> - **Tables 5-7 are missing any parameter count and latency measurements. Why? Please add them where you can. You should also include SOTA (for reference) on the Long Range Arena in Table 5. Likewise for Table 6.**
>
>
>     Table 5
>
>      | Method | IMDB | Cifar10 | Listops | Retrieval | Pathfinder | Complexity |
>     |---|---|---|---|---|---|---|
>     | BigBird   | 64.02 | 40.83 | 36.05 | 59.29 | 74.87 | O(n) |
>     | LongFormer     | 62.85 | 42.22 | 35.63 | 56.89 | 69.71 | O(n) |
>     | Performer    | 65.40 | 42.77 | 18.01 | 53.82 | 77.05 | O(n) |
>     | LinFormer     | 53.94 | 38.56 | 35.70 | 52.27 | 76.34 | O(n) |
>     | Local Attention  | 52.98 | 41.46 | 15.82 | 53.59 | 66.63 | O(n) |
>     | Linear Trans   | 65.90 | 42.34 | 16.13 | 53.09 | 75.30 | O(n) |
>     | S4 | 76.02 | 86.09 | 88.65 | 87.09 | 86.05 | O(n) |
>     | Mega | 90.43 | 90.44 | 63.14 | 91.25 | 96.01 | O(n) |
>     | Sequoia (ours) | 75.10 | 49.88 | 37.70 | 67.04} | 87.30 | O(nlogn) |
>
>
>     Table 6
>
>     | Method | Citeseer | Time | Cora | Time | Pubmed | Time |
>     |---|---|---|---|---|---|---|
>     | TSVM  | 0.640 | -- | 0.575 | -- | 0.622 | -- |
>     | LP | 0.453 | -- | 0.680 | -- | 0.630 | -- |
>     | GRAPHEMB  | 0.432 | -- | 0.672 | -- | 0.653 |
>     | PLANETOID-G  | 0.493 | -- | 0.691 | -- | 0.664 | -- |
>     | PLANETOID-T  | 0.629 | -- | 0.757 | -- | 0.757 | -- |
>     | ChebNet  | 0.698 | -- | 0.780 | -- | 0.744 | -- |
>     | GCN  | 0.703 | 0.25s | 0.815 | 0.3s | 0.790 | 0.32s |
>     | GAT  | 0.725 | 0.8s | 0.830 | 0.88s | 0.790 | 2.42s |
>     | Sequoia (ours) | 0.732 | 0.64s | 0.845 | 0.67s | 0.794 | 2.15s |
>
>
>     Table 7
>
>     | Method | Accuracy | mAcc | Param | Time
>     |---|---|---|---|---|
>     | VoxNet | 85.9 | 83.0 | -- | -- |
>     | MVCNN | 90.1 | -- | -- | -- |
>     | PointNet | 89.2 | 86.0 | 0.6M | 0.81ms |
>     | DGCNN | 92.9 | 90.2 | 1.82M | -- |
>     | GridGCN | 93.1 | 91.3 | -- | 42.2ms |
>     | Set Transformer | 90.4 | -- | -- |
>     | PointNet++ | 91.9 | 88.4 | 1.0M | 1.98ms |
>     | PointConv | 92.5 | -- | -- | -- |
>     | KPConv | 92.9 | -- | 14.3M | 543.7ms |
>     | PointTransformer | 93.7 | 90.6 | 11.7M | 2.0ms |
>     | Sequoia (ours) | 92.0 | 88.4 | 10.9M | 9.6ms |
>
>
>     Table 8
>
>     | Method | Inst IoU | Class IoU | Param | Time
>     |---|---|---|---|---|
>     | PointNet | 71.9 | 43.7 | 3.6M | 1.0 ms|
>     | DGCNN | 85.1 | 82.3 | 1.3M | 125ms |
>     | PointNet++ | 85.1 | 81.9 | 1.0M | 1.41ms |
>     | PointConv | 85.7 | 82.6 | -- |
>     | PointCNN | 86.1 | 84.6 | -- |
>     | PointTransformer | 86.6 | 83.7 | 7.8M | 2.688ms |
>     | Sequoia (ours) | 82.7 | 79.5 | 7.26M | 7.8ms |
>
> - **In Table 10, why does a random hierarchy for 32 clusters perform the best?**
>
>    This may be because when setting the number of clusters equal to 16, this is actually fewer than the actual clusters, limiting the performance of the model. In contrast, when setting the number of clusters to 64, the abundance of allowed clusters makes the model have some unnecessary ones adding noise when processing. However, although 32 is the optimal number of clusters, 64 clusters also yield similar performance, which proves our learnable hierarchical approach is flexible and does not require exhaustive model tuning.
>
> Finally, we hope that we have answered all the concerns and revised our manuscript with all the requests from the reviewer. One more time, we appreciate the reviewer's constructive feedback and suggestions.

---

> > ### Author Response · Authors · 2024-03-12
> > **Visibility issue**
> >
> > Dear Reviewer y1CR,
> >
> > Thank you so much for your review! We sincerely apologize for our mistake. We have answered all your questions in our rebuttal before, but we forgot to set the visibility to you (the rebuttal was only visible to the editor). Please consider our answers and let us know if you need any further response from us! Thank you again!
> >
> > Best,
> > Authors

---

### Review · Reviewer_7yAC · 2024-02-23

**Summary Of Contributions:**

The manuscript proposes a novel architecture that extends self attention transformers by imbuing them with an adaptive hierarchy and redefining the dot product attention operation based on the graph structure of this tree hierarchy. The result is a module that is more efficient over very long sequences and that has inductive biases based on the neighborhood structure of this tree. Lastly, the proposed module combines top-down and bottom-up interactions allowing for more expressive computation.

**Audience:**

Yes

**Broader Impact Concerns:**

There is currently no section about broader impact concerns. I suspect that the hierarchy output by the Cluster Learning Block can help interpret model results and some discussion around this would be beneficial in a broader impact concerns section.

**Claims And Evidence:**

No

**Requested Changes:**

A discussion about the above concerns would be helpful.

**Strengths And Weaknesses:**

### Strengths

Extensive experiments on a large number of datasets and tasks is a key strength of this contribution. This covers domains ranging from images to point clouds to chemistry. Ablation experiments evaluate the impact of varying hyper-parameters.

Computational requirements are measured both in terms of memory footprint and inference wall-clock time. This is important as just measuring FLOPs can be misleading as some sparse operations tend to be very slow on modern accelerators.

------------------------------------------------------------------------------------------------------------------------------
### Weaknesses

The proposed method was very difficult to understand. This was in part due to some undefined terms and in part due to lack of a pseudo-code. The following details some suggested modifications:
- In eq. 1 $\mathcal{A}_{\lambda}$ is undefined and can be confused with the variables in the attention equations on page 6.
- In page 5, section 3.1, last line "we keep the graph unweighted for simplicity". Does this mean that there is no $\varepsilon_ij$?
- In page 7, in the paragraph "Baselines and Implementation Details" it defines number of sub-graphs = 32. "number of sub-graphs" is not defined in the text.
- Section 3.3 was tricky and I think I still do not understand that part. Could the top-down and bottom-up functions be explained using pseudo-code? The same would be very helpful for the cluster learning block and would complement the figures provided.
- Citations could be separated using () to make the text more readable.

The discussion of results is not very precise when commenting on individual tables in the text. In particular,
- In page 8, section 4.1, "we believe that a learnable hierarchy is a main factor that boosts the performance" is not substantially supported by the table. The proposed method could have the top-down and bottom-up processing as the main factor, for example.
- In page 10, section 4.3, "Sequoia is a lightweight attention-based model with competitive results". The results in Table 8 are unfortunately only better than PointNet but worse than all other methods. This needs to be clarified in the text.

----------------------------------------------------------------------------------------------------------------

### Technical concerns

- In section 4.5, is the number of clusters being set per example in the experiments?
- Sparse operations are necessary to realize the attention operations discussed in the manuscript. These normally hurt runtime on GPUs. There is no discussion about this. Did the authors find an efficient implementation?
- The hierarchy is different per example, which means the computation graph has different tensor shapes per example. How does batch computation work in this situation?

---

> ### Author Response · Authors · 2024-03-05
> **Rebuttal for Reviewer 7yAC**
>
> **Weaknesses:**
>
> - **In eq. 1 $\mathcal{A}_{\lambda}$ is undefined and can be confused with the variables in the attention equations on page 6.**
>
>     Thanks for pointing it out; we will clarify the definition. $\mathcal{A}_{\lambda}$ is the adjacency matrix at level $\lambda$.
>
> - **In page 5, section 3.1, last line "we keep the graph unweighted for simplicity". Does this mean that there is no  $\epsilon_{i,j}$ ?**
>
>     We still have $\epsilon_{i,j}$, but it is only binary, determining if two nodes are connected or not without considering the weight of each connection. We will clarify this in the revised paper.
>
> - **In page 7, in the paragraph "Baselines and Implementation Details" it defines number of sub-graphs = 32. "number of sub-graphs" is not defined in the text.**
>
>     "Number of sub-graphs" is the dimension of $P$ and $\Pi$ in section 3.1, Adaptive Hierarchy, which define the max number of clusters the model can propose. We will change this terms to 'max number of clusters', which describes this parameter more accurately.
>
> - **Section 3.3 was tricky and I think I still do not understand that part. Could the top-down and bottom-up functions be explained using pseudo-code? The same would be very helpful for the cluster learning block and would complement the figures provided.**
>
> Thanks for the suggestion; we can add a pseudo-code such as the following:
>
>     Name: Bottom-up
>
>             Input: $X_{lamda - 1}$
>
>             Output: $X_{lamda}$
>
>             $X_{local} = X_{lamda - 1} + MPNN(X_{lamda - 1}))$
>
>             $X_{global} = X_{lamda - 1} + ATTN(X_{lamda - 1}))$
>
>             $X'_{lamda - 1} = X_{local} + X_{global}$
>
>             $X_{lamda} = POOLING(X'_{lamda - 1})$
>
>
>     Name: Top-down
>
>             Input: $X_{lamda - 1}$, $X_{lamda}$
>
>             Output: $X''_{lamda - 1}$
>
>             $X_{local} = X_{lamda} + MPNN(X_{lamda})$
>
>             $X_{global} = X_{lamda} + ATTN(X_{lamda})$
>
>             $X'_{lamda} = X_{local} + X_{global}$
>
>             $X'_{lamda - 1} = UNPOOLING($X'_{lamda})$
>
>             $X''_{lamda - 1} = X_{lamda - 1} + X'_{lamda - 1}$
>
> - **Citations could be separated using () to make the text more readable.**
>
>     We have fixed this problem and will update in our revision.
>
> - **In page 8, section 4.1, "we believe that a learnable hierarchy is a main factor that boosts the performance" is not substantially supported by the table. The proposed method could have the top-down and bottom-up processing as the main factor, for example.**
>
>   We will update the claim. Indeed different ideas within the hierarchical attention are contributing to the performance of the model.
>
> - **In page 10, section 4.3, "Sequoia is a lightweight attention-based model with competitive results". The results in Table 8 are unfortunately only better than PointNet but worse than all other methods. This needs to be clarified in the text.**
>
>
>     | Method | Inst IoU | Class IoU |
>     |---|---|---|
>     | PointNet  | 71.9 | 43.7 |
>     | DGCNN  | 85.1 | 82.3 |
>     | PointNet++  | 85.1 | 81.9 |
>     | PointConv  | 85.7 | 82.6 |
>     | PointCNN  | 86.1 | 84.6 |
>     | PointTransformer  | 86.6 | 83.7 |
>     | Sequoia (ours) - old | 82.7 | 79.5 |
>     | Sequoia (ours) - tuned | 83.8 | 80.6 |
>
>     It is true that for point-cloud tasks, in particular for part segmentation, Sequoia does not seem to be as performant as variety of other tasks and structures that we study in the paper. We will update this claim, as it relates to this  task. We have further improved the results reported in table 8 by hyper-parameter tuning as seen in Table above.
>
>
> **Technical concerns:**
>
> - **In section 4.5, is the number of clusters being set per example in the experiments?**
>
>     Not necessarily, the 'number of clusters' is referred to as the 'number of sub-graphs' in the experiment, which is a hyperparameter of the model which is set based on the dataset rather than per example. We will fix this inconsistency by replacing all terms related to this parameter with 'max number of clusters'.
>
> - **Sparse operations are necessary to realize the attention operations discussed in the manuscript. These normally hurt runtime on GPUs. There is no discussion about this. Did the authors find an efficient implementation?**
>
>     This is efficiently handled by Pytorch Geometric framework which are specialized in Deep Learning for sparse graphs. We will elaborate more implementation details in the our revised manuscript.
>
> - **The hierarchy is different per example, which means the computation graph has different tensor shapes per example. How does batch computation work in this situation?**
>
>     We follow the practice of PyTorch Geometric, which concatenates node and target features in the node dimension and has a separate vector indicating the sample to which each node belongs. This allows samples in a mini-batch to have different sizes.

---

> > ### Comment · Reviewer_7yAC · 2024-03-11
> >
> > The pseudo code is very helpful. Thank you. It explains how $X_{\lambda}$ is computed. How about $\mathcal{E}$?
> >
> > Is ATTN the same as $\mathcal{A}^{ent}$ in page 5? That is it includes attention into ancestor nodes using $A^{a}$?

---

> ### Author Response · Authors · 2024-03-12
> **Answer for question from Reviewer 7yAC**
>
> **The pseudo code is very helpful. Thank you. It explains how $X_\lambda$ is computed. How about $\mathcal{E}$?**
>
> The $\mathcal{E}\_{\lambda}$ is computed when we construct our adaptive hierarchy using the same formula to calculate $\mathcal{A}\_{\lambda}$ (Equation 1, Section 3.1). We will add this to the paper.
>
> $\mathcal{E}\_{\lambda} = \Pi^\top \mathcal{E}\_{\lambda - 1} \Pi.$
>
> This is used in MPNN in the Bottom-up and Top-down Blocks, and we will correct our provided pseudo code from $X_{\text{local}} = X_{\lambda - 1} + MPNN(X_{\lambda - 1})$ to $X_{\text{local}} = X_{\lambda - 1} + MPNN(X_{\lambda - 1}, \mathcal{E}_{\lambda - 1})$."
>
>
> **Is ATTN the same as $\mathcal{A}^\text{ent}$ in page 5? That is it includes attention into ancestor nodes using $\mathcal{A}^a$?**
>
> Yes, they are the same, meaning that the ATTN (i.e. Hierarchical Self-Attention) here includes $\mathcal{A}^a$.

---

> > ### Comment · Reviewer_7yAC · 2024-03-15
> >
> > > Yes, they are the same, meaning that the ATTN (i.e. Hierarchical Self-Attention) here includes $\mathcal{A}^a$.
> >
> > This point confuses me, because:
> > - $X_\lambda$ is required to build the hierarchy,
> > - the hierarchy is required to know which nodes are ancestors,
> > - knowing which nodes are ancestors is required to compute $\mathcal{A}^a$,
> > - $\mathcal{A}^a$ is required to compute $X_\lambda$.
> >
> > This creates a *cyclic* relationship between the quantities meaning you cannot compute the $X_\lambda$ without having the full tree and vice versa. How do you overcome this?

---

> > > ### Author Response · Authors · 2024-03-15
> > > **Answer to an additional question from Reviewer 7yAC**
> > >
> > > **This point confuses me, because:**
> > >
> > > - **$X_\lambda$ is required to build the hierarchy,**
> > > - **the hierarchy is required to know $\mathcal{A}^a$ which nodes are ancestors,**
> > > - **knowing which nodes are ancestors is required to compute,**
> > > - **$X_\lambda$ is required to compute $\mathcal{A}^a$.**
> > > **This creates a cyclic relationship between the quantities meaning you cannot compute the  without having the full tree and vice versa. How do you overcome this?**
> > >
> > > We apologize for the repetition and unclear notation. In fact, in Section 3.1, $\mathcal{A}\_\lambda$ refers to the adjacency matrix, while in Section 3.2 $\mathcal{A}^\text{ent}$ refers to the attention  ATTN in the Bottom-up and Top-down Blocks.
> > >
> > > To clarify this, let me describe the process again: the hierarchy is constructed before we execute the Bottom-up and Top-down Blocks. When building the hierarchy, we only utilize the initial features from $X\_\lambda$ and $\mathcal{A}\_\lambda$ to learn the higher level in hierarchy and apply the pooling equation (Equation 1 in Section 3.1) to obtain the features for $X\_{\lambda+1}$ and $\mathcal{A}\_{\lambda+1}$. During this step, $\mathcal{A}^\text{ent}$, which involves $\mathcal{A}^a$, is not required.
> > >
> > > In the next version, we will use different notations for the two terms for clarity. Thank you again for your feedback!

---

### Author Response · Authors · 2024-03-06
**Summary of Rebuttals**

We would like to express our gratitude to all reviewers once again for their feedback, which significantly benefits our revised paper. Below, we will briefly summarize all the feedback received and our responses:

- Overall, all reviews provided positive feedback regarding the novelty and contribution of our paper, including the valuable idea of leveraging adaptive hierarchy to reduce the complexity of the attention mechanism, as well as the customized attention method that fits this hierarchical structure. Moreover, the reviews also expressed satisfaction with the extensive experiments conducted in our work, which cover a large number of datasets and tasks across various domains, ranging from images to point clouds to chemistry.
- Although we have already provided the time and memory complexity in terms of theory and empirical evidence in some tables of the original manuscript, which was considered one of the strengths of our paper in some reviews, the measurements were actually missing for some datasets. However, we have fully updated these measurements for all datasets as requested by one reviewer.
- The main concern of the reviewers was related to some unclear terms in our paper, but all of these were fortunately addressed in our response, and we will clarify them in our revision. Additionally, the reviewers also pointed out some mistakes related to formats or typos, which can be easily fixed in the next version.

Once again, we sincerely appreciate all the reviewers' comments and will modify our paper according to their suggestions.

---

> ### Comment · Action_Editor_ATVS · 2024-03-18
> **Plan to upload new version?**
>
> Are you planning to upload a revised version of the paper before the reviewer decision recommendations are finalized? This would be useful to verify that major concerns have been addressed.

---

> > ### Author Response · Authors · 2024-03-29
> > **Revision**
> >
> > Dear Editor,
> >
> > Thank you for allowing us to submit the revision! We have just uploaded the revised manuscript with all the updates marked in blue color. We hope that we had addressed all the concerns from reviewers with this revision.
> >
> > Best regards,
> > Authors

---

### Comment · Action_Editor_ATVS · 2024-03-12
**Authors: Please change "Readers" for this rebuttal**

Hi Authors,

Based on the recent comment from Reviewer u9PB, I noticed that your rebuttals to Reviewer's u9PB and y1CR do not include the Reviewers as Readers. Please update the Readers to Everyone and add a comment that you have done so.

Thanks!

---

> ### Author Response · Authors · 2024-03-12
> **Thank you!**
>
> Dear Editor and Reviewers
>
> Thank you so much! We sincerely apologize for this mistake. We have just set the visibility to Everyone. Now, reviewers u9PB and y1CR can see our rebuttals. Please let us know if you need any further response from us! Thank you again!
>
> Best regards,
> Authors

---

### Author Response · Authors · 2024-03-29
**Revision**

Dear Editor and Reviewers,

Thank you very much for your consideration and allowing us to submit the revision! We have just submitted the revision with all the updates (marked in blue color) based on your feedback and recommendations. We hope that our revision will address all of your concerns.

Best regards,
Authors

---

### Decision · Action_Editor_ATVS · 2024-04-02

**Recommendation:** Accept with minor revision

**Comment:**

The paper addresses an important and timely question of interest. All reviewers agreed that the paper is of interest to the TMLR audience, and were generally satisfied with the breadth of the experimental evaluation, along with the inclusion of runtime comparisons. Although reviewers would like to have seen more analysis done on exactly how and why the learned/adaptive hierarchies cluster+organize nodes to improve performance, the method is natural, and it is intuitive why it might work as well as it does.

The biggest reviewer concerns were around the clarity of the presentation. In their revision, alongside new experimental results, the authors added clarifications including updated notation and pseudocode for the core components of the method.

To complete this process, I would like to see the authors add pseudocode for the entire model/method, i.e. that calls the newly added Algorithms 1, 2, and 3 as sub-routines. This should constitute a "minor revision". For example, the following paragraph of the paper describes an implementation of the approach:

> "In graph-level classification, there are 1 layer of message passing to refine the feature before constructing the hierarchy, while in node-level segmentation, we apply four message-passing layers in this process. Thereafter we employ a combination of message-passing, hierarchical attention and residual layers once at each level of the tree to aggregate the global and local information. Additionally, we project the positional encoding (e.g., Laplacian (Dwivedi & Bresson, 2020b) or random walk (Dwivedi et al., 2022a)) to the same dimension of features (d = 96) and add them into the features."

The details of how everything exactly fits together here may be clear/implied to someone closely familiar with s-o-t-a of GNNs, but a more formal description of the implementation, e.g., with pseudo-code could be crucial for future researchers attempting to replicate or build on the work.

Reviewers also asked for details on how the method is efficiently run in practice. In their rebuttal, the authors discussed how they used Pytorch Geometric, including how experimental comparisons are fair w.r.t. the GPU hardware used. These details should be added to the paper, at least in the Appendix.

Beyond this, there are some aesthetic issues that should be corrected in the revision, including the following, some of which were brought up by reviewers:
- inline citation format: use a format that includes parentheses around the citations. This issue is particular evident in the Introduction
- Some tables (2, 3, 4, and 5) are over-wide with the new columns added during the rebuttal. Please modify these so they do not go beyond the text width.
- "Baselines and Implementation Details". Should this just be "Implementation Details"? I don't see any baselines described here.
- "eq. (1)" -> "Eq. 1" and similarly where other Equations are referred to in the text.
- "Pooling eq. (1)" -> "Pooling (Eq. 1)", etc.
- "bottom-up blocks 2" -> "bottom-up blocks (Alg. 2)" and the same where Algorithms 1 and 3 are referred to in the text.
- "we would like to report" -> "we report"
- "there are 1 layer of message passing" -> "there is one layer of message passing"

I encourage the authors to check for other issues of this kind.

**Audience:**

Yes. It's an important and timely topic; the approach is natural and intuitively satisfying.

**Claims And Evidence:**

Yes. All reviewers raised concerns with regards to the clarity of the work. These concerns were addressed by the authors in their rebuttals and revised version of the paper. The presentation could still benefit from additional clarification, but I believe it is in a state where this can be technically accomplished with a minor revision.

---

> ### Author Response · Authors · 2024-04-09
> **Camera-ready version submission**
>
> Dear Editor and Reviewers,
>
> Thank you very much for the opportunity of publication at TMLR! We appreciate your consideration and feedback. We have taken into account all the feedback, comments and suggestions into our camera-ready version.
>
> Best regards,
> Authors